# ENOT: Expectile Regularization for Fast and Accurate Training of Neural Optimal Transport

**Nazar Buzun**[*]
AIRI, MIPT
buzun@airi.net

**Maksim Bobrin**[*]
Skoltech, AIRI
m.bobrin@skoltech.ru

**Dmitry V. Dylov**
Skoltech, AIRI
d.dylov@skoltech.ru

## Abstract

We present a new approach for Neural Optimal Transport (NOT) training procedure, capable of accurately and efficiently estimating optimal transportation plan via specific regularization on dual Kantorovich potentials. The main bottleneck of existing NOT solvers is associated with the procedure of finding a near-exact approximation of the conjugate operator (*i.e.*, the *c*-transform), which is done either by optimizing over non-convex max-min objectives or by the computationally intensive fine-tuning of the initial approximated prediction. We resolve both issues by proposing a new theoretically justified loss in the form of expectile regularization which enforces binding conditions on the learning process of the dual potentials. Such a regularization provides the upper bound estimation over the distribution of possible conjugate potentials and makes the learning stable, completely eliminating the need for additional extensive fine-tuning. Proposed method, called Expectile-Regularized Neural Optimal Transport (ENOT), outperforms previous state-of-the-art approaches in the established Wasserstein-2 benchmark tasks by a large margin (up to a 3-fold improvement in quality and up to a 10-fold improvement in runtime). Moreover, we showcase performance of ENOT for various cost functions in different tasks, such as image generation, demonstrating generalizability and robustness of the proposed algorithm.

*Project page with code*
https://skylooop.github.io/enot/

## 1 Introduction

Computational optimal transport (OT) has enriched machine learning (ML) by offering a new view-angle on the conventional ML tasks through the lens of comparison of probability measures (Villani et al. [2009], Ambrosio et al. [2003], Peyré et al. [2019], Santambrogio [2015]). In different works, OT is primarily employed either 1) as a differentiable proxy, with the OT distance playing the role of a similarity metric between the measures, or 2) as a generative model, defined by the plan of optimal transportation. One notable advantage of using OT in the latter setting is that, compared to other generative approaches, such as GANs, Normalizing Flows, or Diffusion Models, there is no assumption for one of the measures to be defined in a closed form (*e.g.*, Gaussian or uniform) or to be pairwise-aligned, admitting various applications of the OT theory. Both loss objective and generative formulations of OT proved successful in a vast range of modern ML areas, including

---

[*]Equal contribution.

38th Conference on Neural Information Processing Systems (NeurIPS 2024).

generative modelling (Arjovsky et al. [2017], Gulrajani et al. [2017], Korotin et al. [2021], Liu et al. [2019], Leygonie et al. [2019]), reinforcement learning (Fickinger et al. [2022], Haldar et al. [2023], Papagiannis and Li [2022], Luo et al. [2023]), domain adaptation (Xie et al. [2019], Shen et al. [2018]), change point detection (Shvetsov et al. [2020]), barycenter estimation (Kroshnin et al. [2021], Buzun [2023], Bespalov et al. [2022a,b]), biology and genomics (Bunne et al. [2022]). Low dimensional discrete OT problems are solved via Sinkhorn algorithm (Cuturi [2013]), which employs *entropic regularization*. This technique makes the entire optimization problem differentiable and efficient computationally, but may require numerous iterations to converge to an optimal solution, whereas the OT problem for the tasks supported on high-dimensional measure spaces are usually intractable, oftentimes solvable only for the distributions which admit a closed-form density formulations. As a result, the need for computationally-efficient OT solvers has become both evident (Peyré et al. [2019]) and pressing (Montesuma et al. [2023], Khamis et al. [2024]).

In this work, we will be concerned with the complexity, the quality, and the runtime speed of the computational estimation of a *deterministic* OT plan $T$ between two probability measures $\boldsymbol{\alpha}$ and $\boldsymbol{\beta}$ supported on measurable spaces $\mathcal{X}, \mathcal{Y} \subset \mathbb{R}^d$ with Borel sigma-algebra. The OT problem in *Monge's formulation* (MP) for a cost function $c : \mathcal{X} \times \mathcal{Y} \to \mathbb{R}$ is stated as:

$$\mathrm{MP}(\boldsymbol{\alpha}, \boldsymbol{\beta}) = \inf_{T : T_{\#}\boldsymbol{\alpha} = \boldsymbol{\beta}} \int_{\mathcal{X}} c(x, T(x)) \mathrm{d}\boldsymbol{\alpha}(x), \tag{1}$$

where $\{T : T_{\#}\boldsymbol{\alpha} = \boldsymbol{\beta}\}$ is the set of measure-preserving maps, defined by a push forward operator $T_{\#}\boldsymbol{\alpha}(B) = \boldsymbol{\alpha}(T^{-1}(B)) = \boldsymbol{\beta}(B)$ for any Borel subset $B \subset \mathcal{Y}$. The minimizer of the cost above exists if $\mathcal{X}$ is compact, $\boldsymbol{\alpha}$ is *atomless* (i.e. $\forall x \in \mathcal{X} : \boldsymbol{\alpha}(\{x\}) = 0$) and the cost function is continuous (ref. Santambrogio [2015] Theorem 1.22 and Theorem 1.33).

However, MP formulation of the OT problem is intractable, since it requires finding the maps $T$ under the coupling constraints (which is non-convex optimization problem) and is not general enough to provide a way for some mass-splitting solutions. By relaxing constraints in equation (1), the OT problem becomes convex and this form is known as *Kantorovich problem* (KP) (ref. Villani et al. [2009]):

$$\mathrm{KP}(\boldsymbol{\alpha}, \boldsymbol{\beta}) = \inf_{\boldsymbol{\pi} \in \Pi[\boldsymbol{\alpha}, \boldsymbol{\beta}]} \int_{\mathcal{X} \times \mathcal{Y}} c(x, y) d\boldsymbol{\pi}(x, y) = \inf_{\boldsymbol{\pi} \in \Pi[\boldsymbol{\alpha}, \boldsymbol{\beta}]} \mathbb{E}_{\boldsymbol{\pi}}[c(x, y)], \tag{2}$$

where $\Pi[\boldsymbol{\alpha}, \boldsymbol{\beta}] = \{\boldsymbol{\pi} \in \mathcal{P}(\mathcal{X} \times \mathcal{Y}) : \int_{\mathcal{Y}} d\boldsymbol{\pi}(x, y) = d\boldsymbol{\alpha}(x), \int_{\mathcal{X}} d\boldsymbol{\pi}(x, y) = d\boldsymbol{\beta}(y)\}$ is a set of admissible couplings with respective marginals $\boldsymbol{\alpha}, \boldsymbol{\beta}$. MP and KP problems are equivalent in case $\mathcal{X} = \mathcal{Y}$ are compact, the cost function $c(x, y)$ is continuous and $\boldsymbol{\alpha}$ is atomless. Since KP (2) is convex, it admits *dual formulation* (DP), which is constrained concave maximization problem and is derived via Lagrange multipliers (Kantorovich potentials) $f$ and $g$:

$$\mathrm{DP}(\boldsymbol{\alpha}, \boldsymbol{\beta}) = \sup_{(f,g) \in L_1(\boldsymbol{\alpha}) \times L_1(\boldsymbol{\beta})} \left[ \mathbb{E}_{\boldsymbol{\alpha}}[f(x)] + \mathbb{E}_{\boldsymbol{\beta}}[g(y)] \right] + \inf_{\boldsymbol{\pi}, \gamma > 0} \gamma \mathbb{E}_{\boldsymbol{\pi}}[c(x, y) - f(x) - g(y)], \tag{3}$$

where $L_1$ is a set of absolutely integrable functions with respect to underlying measures $\boldsymbol{\alpha}, \boldsymbol{\beta}$. The exchange between infimum and supremum is possible by strong duality (*Slater's* condition). If one decomposes the outer expectation $\mathbb{E}_{\boldsymbol{\pi}}$ in the last equation as $\mathbb{E}_{\boldsymbol{\pi}(x)}\mathbb{E}_{\boldsymbol{\pi}(y|x)}$, we can notice that the supremum by $f(x)$ should satisfy to the condition:

$$f(x) \leq g^c(x) = \inf_{\boldsymbol{\pi}} \mathbb{E}_{\boldsymbol{\pi}(y|x)}[c(x, y) - g(y)] = \inf_{T : \mathcal{X} \to \mathcal{Y}} \left[ c(x, T(x)) - g(T(x)) \right], \tag{4}$$

otherwise, the infimum by $\gamma$ would yield the $-\infty$ value. Operator $g^c$ is called *c-conjugate* transformation. If MP=KP, the solution $\boldsymbol{\pi}(y|x)$ is deterministic and one may set $\boldsymbol{\pi}(y|x) = T(x)$. Finally, DP (3) may be reduced to a single potential optimization task (using inequality (4), ref. Villani et al. [2009] Theorem 5.10):

$$\mathrm{DP}(\boldsymbol{\alpha}, \boldsymbol{\beta}) = \sup_{g \in L_1(\boldsymbol{\beta})} \left[ \mathbb{E}_{\boldsymbol{\alpha}}[g^c(x)] + \mathbb{E}_{\boldsymbol{\beta}}[g(y)] \right] \tag{5}$$

$$= \sup_{g \in L_1(\boldsymbol{\beta})} \inf_{T : \mathcal{X} \to \mathcal{Y}} \left[ \mathbb{E}_{\boldsymbol{\alpha}}[c(x, T(x))] + \mathbb{E}_{\boldsymbol{\beta}}[g(y)] - \mathbb{E}_{\boldsymbol{\alpha}}[g(T(x))] \right]. \tag{6}$$

In practice, during optimization process the infimum by $T$ in c-conjugate transformation is approximated by a parametric model $T_\theta$, such that

$$g^c(x) \leq g^T(x) = c(x, T_\theta(x)) - g(T_\theta(x)). \tag{7}$$

A number of approaches were proposed to model $T_\theta$ in equation (6) *e.g.*, with the Input Convex Neural Networks (ICNN) (Amos et al. [2017], Makkuva et al. [2020], Taghvaei and Jalali [2019]) or with arbitrary non-convex neural networks (Rout et al. [2021], Korotin et al. [2023b]). Most of these approaches make an assumption that the cost is squared Euclidean and utilize Brenier theorem (Brenier [1991]), from which optimal map recovers as gradient of a convex function. The main bottleneck of these parametric solvers is their *instability in finding the optimal c-conjugate potential* $g^T$ from equation (7) and the rough estimation of $T$ often results in a situation where the sum of potentials $g$ and $g^T$ diverges. These instability problems were thoroughly discussed in works Amos [2023], Korotin et al. [2021]. Recently, Amos [2023] showed that it is possible to find a near exact conjugate approximation by performing fine-tuning on top of initial guess ($T_\theta$) in order to achieve closest lower bound in the inequality (7). Despite being an exact approximation to the true conjugate, such procedure requires extensive hyperparameter tuning and will definitely introduce an additional computational overhead.[2]

In this work, we propose to mitigate above issues by constraining the solution class of conjugate potentials through a novel form of *expectile regression* regularization $\mathcal{R}_g$. In order to make joint optimization of $g$ and $T_\theta$ stable and more balanced (optimize $g$ and $T_\theta$ in the OT problem (6) synchronously and with the same frequency), we argue that it is possible to measure proximity of potential $g$ to $(g^T)^c$ without an explicit estimation of the infimum in $c$-conjugate transform (4) and instead optimize the following objective:

$$\mathbb{E}_{\boldsymbol{\alpha}}[g^T(x)] + \mathbb{E}_{\boldsymbol{\beta}}[g(y)] - \mathbb{E}_{\boldsymbol{\alpha},\boldsymbol{\beta}}[\mathcal{R}_g(x,y)]. \tag{8}$$

The regularizer will have to constrain the differences $g(y) - (g^T)^c(y)$ and $g^T(x) - g^c(x)$ that should match at the end of training. We show that such a natural regularization outperforms the state-of-the-art NOT approaches in all of the tasks of the established benchmark for the computational OT problems (the Wasserstein-2 benchmark, presented in Korotin et al. [2021]), with a remarkable 5 to 10-fold acceleration of training compared to previous works and achieving faster convergence on synthetic datasets with desirable properties posed on OT map. Moreover, we show that proposed method obtains state-of-the-art results on generative image-to-image tasks in terms of FID and MSE.

## 2 Related Work

In the essence, the main challenge of finding the optimal Kantorovich potentials in equation (6) lies in alternating computation of the exact $c$-conjugate operators (4). Recent approaches consider the dual OT problem from the perspective of optimization over the parametrized family of potentials. Namely, parametrizing potential $g_\eta$ either as a non-convex Multi-Layer Perceptron (MLP) (Dam et al. [2019]) or as an Input-Convex Neural Network (ICNN) (Amos et al. [2017]). Different strategies for finding the solution to the conjugate operator can be investigated under a more general formulation of the following optimization (Makkuva et al. [2020], Amos [2023]):

$$\max_\eta \Big[ - \mathbb{E}_{\boldsymbol{\alpha}}[g_\eta(\widehat{T}(x))] + \mathbb{E}_{\boldsymbol{\beta}}[g_\eta(y)] \Big], \quad \min_\theta \mathbb{E}_{\boldsymbol{\alpha}} \Big[ \mathcal{L}_{\mathrm{amor}}(T_\theta(x), \widehat{T}(x)) \Big], \tag{9}$$

with $\widehat{T}(x)$ being the fine-tuned argmin of $c$-conjugate transform (4) with initial value $T_\theta(x)$. Loss objective $\mathcal{L}_{\mathrm{amor}}$ can be one of three types of amortization losses which makes $T_\theta(x)$ converge to $\widehat{T}(x)$. This max-min problem is similar to adversarial learning, where $g_\eta$ acts as a discriminator and $T_\theta$ finds a deterministic mapping from the measure $\boldsymbol{\alpha}$ to $\boldsymbol{\beta}$. The first objective in equation (9) is well-defined under certain assumptions and the optimal parameters can be found by differentiating w.r.t. $\eta$, according to the Danskin's envelope theorem (ref. Danskin [1966]). We briefly overview main design choices of the amortized models $T_\theta(x)$ in the form of continuous dual solvers and the corresponding amortization objective options for $\mathcal{L}_{\mathrm{amor}}$ in the Appendix C.

Another method considers the solution to the optimal map in (1) from a different perspective by introducing a regularization term named Monge Gap (Uscidda and Cuturi [2023]) and learns optimal $T$ map from Monge formulation directly without any dependence on conjugate potentials. More explicitly, by finding the reference measure $\boldsymbol{\mu}$ with Support($\boldsymbol{\alpha}$) $\subset$ Support($\boldsymbol{\mu}$), the following regularizer quantifies deviation of $T$ from being optimal transport map:

$$\mathcal{M}^c_{\boldsymbol{\mu}} = \mathbb{E}_{\boldsymbol{\mu}}[c(x, T(x))] - \mathrm{KP}^\varepsilon(\boldsymbol{\mu}, T_\#\boldsymbol{\mu}) \tag{10}$$

---

[2]This intuition is supported by a direct evaluation in Section 5 below.

with KP$^\varepsilon$ being entropy-regularized Kantorovich problem (2). However, despite its elegance, we still need some method to compute KP$^\varepsilon$, and the underlying measure $\boldsymbol{\mu}$ should be chosen thoughtfully, considering that its choice impacts the resulting optimal transport map and the case when $\boldsymbol{\mu} = \boldsymbol{\alpha}$ does not always provide expected outcomes.

## 3 Background

**Bidirectional transport mapping.** We employ notations with hats $(\widehat{\boldsymbol{\pi}}, \widehat{T}, \widehat{g}, \widehat{f} = (\widehat{g})^c)$ to indicate the correspondence to the solution (argmins) of the OT problem (6). Optimality in equation (6) is obtained whenever complementary slackness is satisfied, namely: $\forall (x, y) \in \text{Support}(\widehat{\boldsymbol{\pi}}) : \widehat{g}^c(x) + \widehat{g}(y) = c(x, y)$. Consider a specific setting when the optimal transport plan $\widehat{\boldsymbol{\pi}}(x, y)$ is deterministic and MP=KP. Let the domains of $\boldsymbol{\alpha}, \boldsymbol{\beta}$ be equal and compact, i.e. $\mathcal{X} = \mathcal{Y}$, for some strictly convex function $h$ the cost $c(x, y) = h(x - y)$. Denote by $h^*$ the convex conjugate of $h$, implying that $(\partial h)^{-1} = \nabla h^*$. If $\boldsymbol{\alpha}$ is absolutely continuous, then $\widehat{\boldsymbol{\pi}}(x, y)$ is unique and concentrated on graph $(x, \widehat{T}(x))$. Moreover, one may link it with Kantorovich potential $\widehat{f}$ as follows (Santambrogio [2015] Theorem 1.17): $\nabla \widehat{f}(x) \in \partial_x c(x, \widehat{T}(x))$ and particularly for $c(x, y) = h(x - y)$

$$\widehat{T}(x) = x - \nabla h^*(\nabla \widehat{f}(x)). \tag{11}$$

If the same conditions are met for measure $\boldsymbol{\beta}$ we can express the inverse mapping $\widehat{T}^{-1}(y)$ through the potential $\widehat{g}$:

$$\widehat{T}^{-1}(y) = y - \nabla h^*(\nabla \widehat{g}(y)). \tag{12}$$

Max-min optimization in problem (6) by means of parametric models $f_\theta$ and $g_\eta$ is unstable due to non-convex nature of the problem. One way to improve robustness is to simultaneously train bidirectional mappings $\widehat{T}(x)$ and $\widehat{T}^{-1}(y)$ expressed by formulas (11) and (12), thus yielding self-improving iterative procedure. During the training, we also may use the equations (11) and (12) with non-optimal functions $f_\theta$ and $g_\eta$, because there are no restrictions on $T$ in problem (6) and we can use any representation for the transport mapping function. Under weaker constraints (for example $\mathcal{X} = \mathcal{Y} = \mathbb{R}^d$), the c-concavity of the potentials $f$ and $g$ may be required. In this case, we can rely on a local c-concavity in the data concentration region or, if the conditions for equations (11) and (12) are not satisfied, we can use an arbitrary function for $T_\theta$ and do not express it through the potential $f_\theta$. Under Brenier's theorem conditions (Brenier [1991]) in domain $\mathbb{R}^n$ for the squared Euclidean cost, it holds that $\widehat{T}(x) = x - \nabla \widehat{f}(x)$, where $\widehat{f}$ is some $l_2$-concave function. It follows that the optimal potentials $\widehat{f}$ and $\widehat{g}$ are $l_2$-concave, even if one uses not $l_2$-concave potentials $f, g$ in the training process.

**Expectile regression.** The idea behind the proposed regularization approach is to minimize the least asymmetrically weighted squares. It is a popular option for estimating conditional maximum of a distribution through neural networks. Recently, expectile regression was used in some offline Reinforcement Learning algorithms and representation learning approaches (Ghosh et al. [2023]). Let $f_\theta : \mathbb{R}^d \to \mathbb{R}$ be some parametric model from $L_2(\mathbb{R}^d)$ space and $x, y$ be dependent random variables in $\mathbb{R}^d \times \mathbb{R}$, where $y$ has finite second moment. By definition (Newey and Powell [1987]), the expectile regression problem is:

$$\min_\theta \mathbb{E}\Big[\mathcal{L}_\tau(y - f_\theta(x))\Big] = \min_\theta \mathbb{E}\big|\tau - \mathbb{I}[y \le f_\theta(x)]\big| (y - f_\theta(x))^2, \quad \tau \ge 0.5. \tag{13}$$

The expectation is taken over the $\{x, y\}$ pairs. The asymmetric loss $\mathcal{L}_\tau$ reduces the contribution of those values of $y$ that are smaller than $f_\theta(x)$, while the larger values are weighted more heavily (ref. Figure 5). The expectile model $f_\theta(x)$ is strictly monotonic in parameter $\tau$. Particullary, the important property for us is when $\tau \to 1$, it approximates the conditional (on $x$) maximum operator over the corresponding values of $y$ (Bellini et al. [2014]). Below we compute c-conjugate transformation by means of expectile.

## 4 Proposed Method

The main motivation behind our method is to regularize optimization objective in DP (6) with non-exact approximation of $c$-conjugate potential $g^T(x)$, defined in (7). The regularisation term

$\mathcal{R}_g(x, y)$ should "pull" $g(y)$ towards $(g^T)^c(y)$ and $g^T(x)$ towards $g^c(x)$. Instead of finding explicit $c$-conjugate transform, we compute $\tau$-expectile of random variables $g^T(x) - c(x, y)$, treating $y$ as a condition. From the properties of expectile regression described above and equation (4) follows that when $\tau \to 1$, the expectile converges to

$$\max_{x \in \mathcal{X}} \left[ g^T(x) - c(x, y) \right] = -(g^T)^c(y). \tag{14}$$

Let the parametric models of Kantorovich potentials be represented as $f_\theta(x)$ and $g_\eta(y)$. The transport mapping $T_\theta(x)$ has the same parameters as $f_\theta(x)$ if it can be expressed through $f_\theta$ (ref. 11), or otherwise, when $f_\theta$ is not used (one-directional training), it is its own parameters. Let approximate the maximum of eq. (14) by $\tau$-expectile of $g_\eta^T(x) - c(x, y)$ conditioning on $y$. So the target (term $y$ in eq. 13) of the expectile regression here is $g_\eta^T(x) - c(x, y)$. The model in this case is $-g_\eta(y)$. It has a negative sign because we approximate c-transform of $g_\eta^T(x)$, which equals to $\inf_x c(x, y) - g_\eta^T(x)$. The corresponding regression loss is $\mathcal{L}_\tau\big(g_\eta^T(x) - c(x, y) + g_\eta(y)\big)$. Accounting the definition of $g^T$ (7) we obtain the regularization loss for potential $g_\eta$:

$$\mathcal{R}_g(\eta, x, y) = \mathcal{L}_\tau\big(c(x, T_\theta(x)) - g_\eta(T_\theta(x)) - c(x, y) + g_\eta(y)\big). \tag{15}$$

The proposed expectile regularisation is incorporated into alternating step of learning the Kantorovich potentials by implicitly estimating $c$-conjugate transformation, additionally encouraging model $g$ to satisfy the *c-concavity criterion* (Villani et al. [2009] Proposition 5.8). We minimize $R_g(\eta) = \mathbb{E}_{\boldsymbol{\alpha}, \boldsymbol{\beta}} \mathcal{R}_g(\eta, x, y)$ by $\eta$ and simultaneously do training of the dual OT problem (6), splitting it into two losses

$$L_g(\eta) = -\mathbb{E}_{\boldsymbol{\beta}}[g_\eta(y)] + \mathbb{E}_{\boldsymbol{\alpha}}[g_\eta(T_\theta(x))], \tag{16}$$

$$L_f(\theta) = -\mathbb{E}_{\boldsymbol{\alpha}}[g_\eta(T_\theta(x))] + \mathbb{E}_{\boldsymbol{\alpha}}[c(x, T_\theta(x))]. \tag{17}$$

---

**Algorithm 1** ENOT Training

---

**Input**: samples from unknown distributions $x \sim \boldsymbol{\alpha}$ and $y \sim \boldsymbol{\beta}$; cost function $c(x, y)$;
**Parameters**: parametric potential model `f` or vector field `f = T`, parametric potential model `g`, optimizers `opt_f` and `opt_g`, batch size $n$, train steps $N$, expectile $\tau$, expectile loss weight $\lambda$, bidirectional training flag `is_bidirectional`;
**function** `train_step(f, g,` $\{x_1, \ldots, x_n\}, \{y_1, \ldots, y_n\})$
 1: {Assign OT mapping $\texttt{T}(x)$}
 2: **if** `is_bidirectional` is **true** $\texttt{T}(x) = x - \nabla h^*(\nabla \texttt{f}(x))$ **else** $\texttt{T}(x) = \texttt{f}(x)$
 3: {Compute dual OT losses and expectile regularisation $R_\texttt{g}$}
 4: $L_\texttt{g} = \frac{1}{n} \sum_{i=1}^n \texttt{g}(\texttt{T}(x_i)) - \frac{1}{n} \sum_{i=1}^n \texttt{g}(y_i)$
 5: $L_\texttt{f} = \frac{1}{n} \sum_{i=1}^n \big[c(x_i, \texttt{T}(x_i)) - \texttt{g}(\texttt{T}(x_i))\big]$
 6: $R_\texttt{g} = \frac{1}{n} \sum_{i=1}^n \mathcal{L}_\tau\big(c(x_i, \texttt{T}(x_i)) - c(x_i, y_i) + \texttt{g}(y_i) - \texttt{g}(\texttt{T}(x_i))\big)$
 7: {Apply gradient updates for parameters of models `f` and `g`}
 8: `opt_f.minimize(f,` $\text{loss} = L_\texttt{f}$`)`
 9: `opt_g.minimize(g,` $\text{loss} = L_\texttt{g} + \lambda R_\texttt{g}$`)`
**end function**
10: {Main train loop}
11: **for** $t \in 1, \ldots, N$ **do**
12:     sample $x_1, \ldots, x_n \sim \boldsymbol{\alpha}, \; y_1, \ldots, y_n \sim \boldsymbol{\beta}$
13:     **if** `is_bidirectional` is **false or** $t \mod 2 = 0$ **then**
14:        `train_step(f, g,` $\{x_1, \ldots, x_n\}, \{y_1, \ldots, y_n\})$
15:     **else**
16:        {Update inverse mapping $\boldsymbol{\beta} \to \boldsymbol{\alpha}$ by swapping `f` and `g`}
17:        `train_step(g, f,` $\{y_1, \ldots, y_n\}, \{x_1, \ldots, x_n\})$
18:     **end if**
19: **end for**
20: sample $x_1, \ldots, x_n \sim \boldsymbol{\alpha}, \; y_1, \ldots, y_n \sim \boldsymbol{\beta}$
21: {Approximate OT distance by sum of conjugate potentials, get $T$ from step 2}
22: `dist` $= \frac{1}{n} \sum_{i=1}^n \big[\texttt{g}(y_i) + c(x_i, \texttt{T}(x_i)) - \texttt{g}(\texttt{T}(x_i))\big]$; $\texttt{f}(x) = c(x, \texttt{T}(x)) - \texttt{g}(\texttt{T}(x))$
23: **return** `f, g, dist`

---

Algorithm 1 describes a complete training loop with full objective expression $L_f(\theta) + L_g(\eta) + \lambda R_g(\eta)$ and hyperparameters $\tau$ (expectile) and $\lambda$. It includes two training options: one-directional with models $g_\eta$ and $T_\theta$; and bidirectional for strictly convex cost functions in form of $h(x - y)$ with models $f_\theta, g_\eta$ and $T_\theta, T_\eta^{-1}$ (the latter are represented in terms of $f_\theta, g_\eta$ by formulas (11), (12)). The bidirectional training procedure updates $g_\eta, T_\theta$ in one optimization step and then switches to $f_\theta, T_\eta^{-1}$ update in the next step. This option includes analogical regularisation term for the potential $f_\theta$:

$$\mathcal{R}_f(\theta, x, y) = \mathcal{L}_\tau\big(c(T_\eta^{-1}(y), y) - f_\theta(T_\eta^{-1}(y)) - c(x, y) + f_\theta(x)\big). \tag{18}$$

In the end of training we approximate the correspondent Wasserstein distance by expression

$$W_c(\boldsymbol{\alpha}, \boldsymbol{\beta}) = \mathbb{E}_{\boldsymbol{\beta}}[g_{\widehat{\eta}}(y)] - \mathbb{E}_{\boldsymbol{\alpha}}[g_{\widehat{\eta}}(T_{\widehat{\theta}}(x))] + \mathbb{E}_{\boldsymbol{\alpha}}[c(x, T_{\widehat{\theta}}(x))] \tag{19}$$

with optimized parameters $\widehat{\theta}, \widehat{\eta}$. We include a formal convergence analysis for $\tau$ regularized functions being a tight bound on the exact solution to the c-conjugate transform in Appendix D.

## 5 Experiments

In this section, we provide a thorough validation of ENOT on a popular W2 benchmark to test the quality of recovered OT maps. We compare ENOT with the state-of-the-art approaches and also showcase its performance in generative tasks. Additional results and visualizations on 2D *synthetic* tasks are provided in Appendix F.

### 5.1 Results on Wasserstein-2 Benchmark

While evaluating ENOT, we also measured the wall-clock runtime on all Wasserstein-2 benchmark data (Korotin et al. [2021]). The tasks in the benchmark consist of finding the optimal map under the squared Euclidean norm $c(x, y) = \|x - y\|^2$ between either: (**1**) high-dimensional (HD) pairs $(\boldsymbol{\alpha}, \boldsymbol{\beta})$ of Gaussian mixtures, where the target measure is constructed as an average of gradients of learned ICNN models via $W_2$(Korotin et al. [2019]) or (**2**) samples from pretrained generative model W2GN (Korotin et al. [2021]) on CelebA dataset (Liu et al. [2015]). The quality of the map $\widehat{T}$ from $\boldsymbol{\alpha}$ to $\boldsymbol{\beta}$ is evaluated against the ground truth optimal transport plan $T^*$ via *unexplained variance percentage* metric ($\mathcal{L}_2^{\mathrm{UV}}$) (Korotin et al. [2019, 2021], Makkuva et al. [2020]), which quantifies deviation from the optimal alignment $T^*$, normalized by the variance of $\boldsymbol{\beta}$:

$$\mathcal{L}_2^{\mathrm{UV}}(\widehat{T}, \boldsymbol{\alpha}, \boldsymbol{\beta}) = 100 \cdot \frac{\mathbb{E}_{\boldsymbol{\alpha}}\|\widehat{T}(x) - T^*(x)\|^2}{\mathrm{Var}_{\boldsymbol{\beta}}[y]}. \tag{20}$$

The results of the experiments are provided in Table 1 for CelebA64 ($64 \times 64$ image size) and in Table 2 for the mixture of Gaussian distributions with a varying number of dimensions $D$. Overall, ENOT manages to approximate optimal plan $T^*$ accurately and without any computational overhead compared to the baseline methods which require an inner conjugate optimization loop solution. To be consistent with the baseline approaches, we averaged our results across 3-5 different seeds. All the hyperparameters are listed in Appendix E.2 (Table 6).

Despite the fact that we compute at each train step a *non-exact* c-transform, the expectile regularization enables the method to outperform all *exact* methods in all our extensive tests. In actuality, the regularization does not introduce an additional bias, neither in theory, nor in practice. At the end of training (or upon convergence), we obtain the exact estimate of the c-conjugate transformation. Other methods demand near-exact estimation at each optimization step, requiring additional inner optimization and introducing significant overhead. We assume that introduces an imbalance in the simultaneous optimization by $g$ and $T$ in equation (6), underestimating the OT distance as a result.

### 5.2 Different Cost Functionals

We further investigate how ENOT performs for different cost functions and compare Monge gap regularization (Uscidda and Cuturi [2023]) and ENOT between the measures defined on 2D synthetic datasets. In Figure 1, we observe that despite recovering Monge-like transport maps $T_\theta$, ENOT achieves convergence up to $2\times$ faster and produces more desirable OT-like optimal maps. To test other specific use cases, we conducted experiments on 2D spheres data (Figure 2), where we parametrize

| Method | Conjugate | Early Generator | Mid Generator | Late Generator |
|--------|-----------|-----------------|---------------|----------------|
| W2-Cycle | None | 1.7 | 0.5 | 0.25 |
| MM-Objective | None | 2.2 | 0.9 | 0.53 |
| MM-R-Objective | None | 1.4 | 0.4 | 0.22 |
| W2OT-Cycle | None | $> 100$ | $26.50 \pm 60.14$ | $0.29 \pm 0.59$ |
| W2OT-Objective | None | $> 100$ | $0.29 \pm 0.15$ | $0.69 \pm 0.9$ |
| W2OT-Cycle | L-BFGS | $0.62 \pm 0.01$ | $0.20 \pm 0.00$ | $0.09 \pm 0.00$ |
| W2OT-Objective | L-BFGS | $0.61 \pm 0.01$ | $0.20 \pm 0.00$ | $0.09 \pm 0.00$ |
| W2OT-Regression | L-BFGS | $0.62 \pm 0.01$ | $0.20 \pm 0.00$ | $0.09 \pm 0.00$ |
| W2OT-Cycle | Adam | $0.65 \pm 0.02$ | $0.21 \pm 0.00$ | $0.11 \pm 0.05$ |
| W2OT-Objective | Adam | $0.65 \pm 0.02$ | $0.21 \pm 0.00$ | $0.11 \pm 0.05$ |
| W2OT-Regression | Adam | $0.66 \pm 0.01$ | $0.21 \pm 0.00$ | $0.12 \pm 0.00$ |
| **ENOT (Ours)** | None | $0.32 \pm 0.011$ | $0.08 \pm 0.004$ | $0.04 \pm 0.002$ |

Table 1: $\mathcal{L}_2^{\mathrm{UV}}$ comparison of ENOT on CelebA64 tasks from the Wasserstein-2 benchmark. The attributes after the method names ('Cycle', 'Objective', 'Regression') correspond to the type of amortisation loss. Column 'Conjugate' indicates the selected optimizer for the internal fine-tuning of $c$-conjugate transform. The results of our method include the mean and the standard deviation across 3 different seeds. The best scores are highlighted.

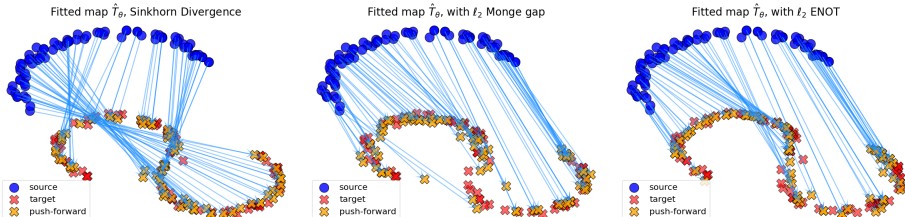

Figure 1: Fitting of three different transport maps $T_\theta$ between source and target measures in $\mathbb{R}^2$ with Euclidean cost function $c(x, y) = \|x - y\|$. We use the same number of iterations and MLP architecture for each method. **Left**: Sinkhorn divergence; **Middle**: Monge gap; **Right**: ENOT.

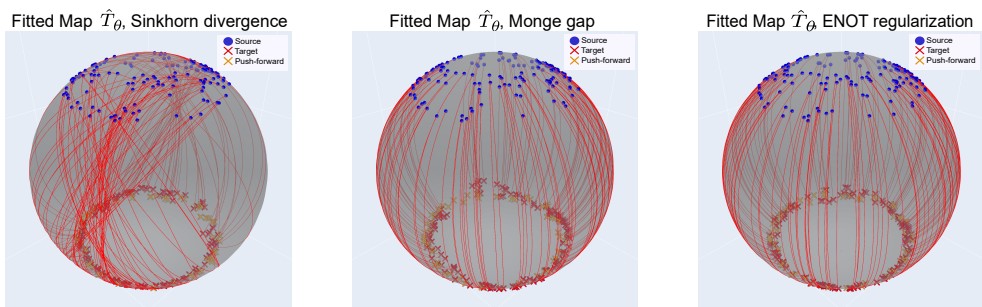

Figure 2: Recovered OT maps $T_\theta$ between synthetic measures on 2-sphere with geodesic cost $c(x, y) = \arccos(x^T y)$. All models are MLPs with outputs normalized to be on a unit sphere. Blue dots are the empirical source measure, red crosses are the empirical target measure and the orange crosses are the result of the found transport map. **Left**: Sinkhorn; **Middle**: Monge; **Right**: ENOT.

the map $T_\theta$ as a `MLP` and test the algorithms with the geodesic cost $c(x, y) = \arccos(x^T y)$ with $n = 1000$ iterations. We set here flag `is_bidirectional=False` (meaning the training mode is one-directional in this example). Remarkably, the time required for convergence is minimal for ENOT, while the Monge gap takes up to three times longer. Moreover, in our experiments, Monge gap solver diverged for $n > 1300$ iterations. ENOT consistently estimates accurate and continuous OT maps.

| Method | Conjugate | $D = 2$ | $D = 4$ | $D = 8$ | $D = 16$ | $D = 32$ |
|---|---|---|---|---|---|---|
| W2-Cycle | None | 0.1 | 0.7 | 2.6 | 3.3 | 6.0 |
| MM-Objective | None | 0.2 | 1.0 | 1.8 | 1.4 | 6.9 |
| MM-R-Objective | None | 0.1 | 0.68 | 2.2 | 3.1 | 5.3 |
| Monge Gap | None | $0.1 \pm 0.0$ | $0.57 \pm 0.0$ | $2.05 \pm 0.06$ | $4.22 \pm 0.1$ | $7.24 \pm 0.17$ |
| W2OT-Cycle | None | $0.05 \pm 0.0$ | $0.35 \pm 0.01$ | $> 100$ | $> 100$ | $> 100$ |
| W2OT-Objective | None | $> 100$ | $> 100$ | $> 100$ | $> 100$ | $> 100$ |
| W2OT-Cycle | L-BFGS | $> 100$ | $> 100$ | $> 100$ | $> 100$ | $> 100$ |
| W2OT-Objective | L-BFGS | $0.03 \pm 0.0$ | $0.22 \pm 0.01$ | $0.6 \pm 0.03$ | $0.8 \pm 0.11$ | $2.09 \pm 0.31$ |
| W2OT-Regression | L-BFGS | $0.03 \pm 0.0$ | $0.22 \pm 0.01$ | $0.61 \pm 0.04$ | $0.77 \pm 0.1$ | $1.97 \pm 0.38$ |
| W2OT-Cycle | Adam | $0.18 \pm 0.03$ | $0.69 \pm 0.56$ | $1.62 \pm 2.82$ | $> 100$ | $> 100$ |
| W2OT-Objective | Adam | $0.06 \pm 0.01$ | $0.26 \pm 0.02$ | $0.63 \pm 0.07$ | $0.81 \pm 0.10$ | $1.99 \pm 0.32$ |
| W2OT-Regression | Adam | $0.22 \pm 0.01$ | $0.28 \pm 0.02$ | $0.61 \pm 0.07$ | $0.8 \pm 0.10$ | $2.07 \pm 0.38$ |
| **ENOT (Ours)** | None | $0.02 \pm 0.0$ | $0.03 \pm 0.001$ | $0.14 \pm 0.01$ | $0.24 \pm 0.03$ | $0.67 \pm 0.02$ |

| Method | Conjugate | $D = 64$ | $D = 128$ | $D = 256$ |
|---|---|---|---|---|
| W2-Cycle | None | 7.2 | 2.0 | 2.7 |
| MM-Objective | None | 8.1 | 2.2 | 2.6 |
| MM-R-Objective | None | 10.1 | 3.2 | 2.7 |
| Monge Gap | None | $7.99 \pm 0.19$ | $9.1 \pm 0.29$ | $9.41 \pm 0.21$ |
| W2OT-Cycle | None | $> 100$ | $> 100$ | $> 100$ |
| W2OT-Objective | None | $> 100$ | $> 100$ | $> 100$ |
| W2OT-Cycle | L-BFGS | $> 100$ | $> 100$ | $> 100$ |
| W2OT-Objective | L-BFGS | $2.08 \pm 0.40$ | $0.67 \pm 0.05$ | $0.59 \pm 0.04$ |
| W2OT-Regression | L-BFGS | $2.08 \pm 0.39$ | $0.67 \pm 0.05$ | $0.65 \pm 0.07$ |
| W2OT-Cycle | Adam | $> 100$ | $> 100$ | $> 100$ |
| W2OT-Objective | Adam | $2.21 \pm 0.32$ | $0.77 \pm 0.05$ | $0.66 \pm 0.07$ |
| W2OT-Regression | Adam | $2.37 \pm 0.46$ | $0.77 \pm 0.06$ | $0.75 \pm 0.09$ |
| **ENOT (Ours)** | None | $0.56 \pm 0.03$ | $0.3 \pm 0.01$ | $0.51 \pm 0.02$ |

Table 2: $\mathcal{L}_2^{\mathrm{UV}}$ comparison of ENOT with baseline methods on the high-dimensional (HD) tasks from Wasserstein-2 benchmark. The suffixes ('Cycle', 'Objective', 'Regression') correspond to the type of amortisation loss. Column 'Conjugate' indicates the selected optimizer for the internal fine-tuning of $c$-conjugate transform. $D$ is the dimension of the measures domain. The mean and the standard deviations of our method are computed across 5 different seeds. The best scores are highlighted.

## 5.3 Unpaired Image-to-Image Translation

To showcase the power of expectile regularization beyond the $W_2$ benchmarks, we apply our method to an unpaired image-to-image translation task. The corresponding image datasets are: female subset of Celebrity faces (CelebA(f)) (Liu et al. [2015]), Anime Faces (Anime)[3], Flickr-Faces-HQ (FFHQ) (Karras et al. [2019]), comic faces v2 (Comics)[4], Handbags and Shoes[5]. The datasets are pre-processed in the conventional way as described in (Gazdieva et al. [2023]). The trained transport maps include: Handbags to Shoes, FFHQ to Comics, CelebA(f) to Anime. We employ squared Euclidean cost function divided by the image size (64 or 128), basic U-Net architecture (Ronneberger et al. [2015]) for the transport map $T_\theta(x)$, and ResNet from WGAN-QC (Liu et al. [2019]) as a potential $g_\eta(y)$. ENOT trains in one-directional mode with total steps count $N = 120k$. Appendix Table 7 contains a complete list of hyperparameters (conventionally, we have used FID (Heusel et al. [2017]) metric for the hyperparameters tuning). We report the learned transport maps in Figure 3 as well as the widely used FID and MSE metrics in Table 3. Appendix 10 includes additional evaluation on test images.

---

[3] kaggle.com/datasets/reitanaka/alignedanimefaces
[4] kaggle.com/datasets/defileroff/comic-faces-paired-synthetic-v2
[5] github.com/junyanz/iGAN/blob/master/train_dcgan

Image-to-image translation baselines include popular GAN-based approaches: CycleGAN (Zhu et al. [2017]) and StarGAN-v2 (Choi et al. [2020]), and two other recent neural OT methods: Extremal OT (Gazdieva et al. [2023]) and Kernel OT (Korotin et al. [2023a]). *ENOT outperforms the baselines in all tasks in therms of FID score and, as in all other experiments, significantly speeds up the computation.* It takes about 5 hours to train the transport model on one GPU RTX 3090 Ti with image size $64 \times 64$ and about 16 hours when the image size is $128 \times 128$, while an approximate training time of the other OT algorithms and GANs is about 3 days on the same GPU.

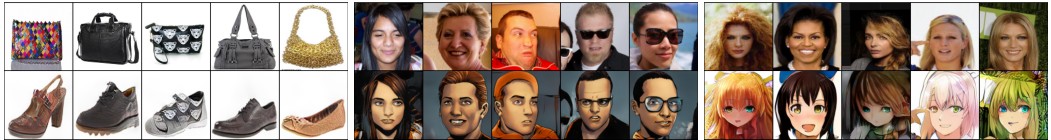

Figure 3: **Left**: Handbags to Shoes; **Middle**: FFHQ to Comics; **Right**: CelebA(f) to Anime; all images sizes are 128x128, the 1st row contains the source images, the 2nd row contains predicted generative mapping by ENOT; **Cost function**: $L^2$ divided by the image size.

| Task and image size | CycleGAN | StarGAN | Extr. OT | Ker. OT | ENOT |
|---|---|---|---|---|---|
| Handbags $\Rightarrow$ Shoes 128 | 23.4 | 22.36 | 27.10 | 26.7 | 19.19 |
| FFHQ $\Rightarrow$ Comics 128 | - | - | 20.95 | 20.81 | 17.11 |
| CelebA(f) $\Rightarrow$ Anime 64 | 20.8 | 22.40 | 14.65 | 18.28 | 13.12 |
| CelebA(f) $\Rightarrow$ Anime 128 | - | - | 19.44 | 21.96 | 18.85 |

FID Metric

| Task and image size | CycleGAN | StarGAN | Extr. OT | Ker. OT | ENOT |
|---|---|---|---|---|---|
| Handbags $\Rightarrow$ Shoes 128 | 0.43 | 0.24 | 0.37 | 0.37 | 0.34 |
| FFHQ $\Rightarrow$ Comics 128 | - | - | 0.22 | 0.21 | 0.20 |
| CelebA(f) $\Rightarrow$ Anime 64 | 0.32 | 0.21 | 0.30 | 0.34 | 0.26 |
| CelebA(f) $\Rightarrow$ Anime 128 | - | - | 0.31 | 0.36 | 0.28 |

MSE Metric

Table 3: Comparison of ENOT to baseline methods for image-to-image translation. We evaluate generation task between two different datasets: Source $\Rightarrow$ Target. And compare resulting images based on Frechet Inception Distance (FID) and Mean Squared Error (MSE). Empty cells indicate that original authors of particular method did not include results for those tasks.

## 5.4 Ablation Study: Varying hyperparameters expectile and regularization weight

Figure 4 presents the study of the impact of the proposed expectile regularization on the $\mathcal{L}_2^{UV}$ metric. This is done by varying the values of the expectile hyperparameter $\tau$ and the scaling of the expectile loss coefficient $\lambda$ in Algorithm 1. Colored contour plots show the areas of the lowest and the highest values of $\mathcal{L}_2^{UV}$. The grey areas depict the cases when the OT solver diverged. For example, in high-dimensions, $D \geq 64$, it is the case for $\lambda = 0$, pointing out that the expectile regularization with $\tau$ is necessary to prevent the instability during training.

The ablation study shows that, even when the parameter choices of $\tau$ and $\lambda$ are not optimal, *ENOT still outperforms the other baseline solvers* in Table 2, making ENOT approach robust to extensive hyperparameter tuning, compared to amortized optimization approach (Amos [2023]) sensitive to the hyperparameters of the conjugate solver. All ablation study experiments were conducted using the network structure and the learning rates in Appendix E.2 (Table 4) (they coincide with those in Table 2).

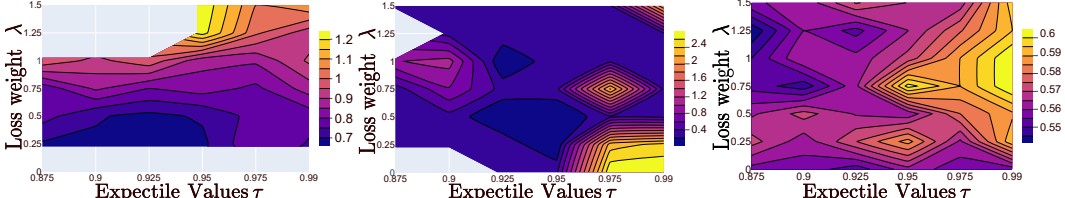

Figure 4: Contour plots of $\mathcal{L}_2^{\text{UV}}$ dependence on the values of $\lambda$ and $\tau$ in Algorithm 1 for the dimensions of $D = 256$ (Left, NaN values are greyed out), $D = 128$ (Middle), and $D = 64$ (Right).

## 6 Conclusion, Limitations and Future Work

Our paper introduces a new method, ENOT, for efficient computation of the conjugate potentials in neural optimal transport with the help of expectile regularisation. We show that a solution to such a regularization objective is indeed a close approximation to the true $c$-conjugate potential. Remarkably, ENOT surpasses the current state-of-the-art approaches, yielding an up to a 10-fold improvement in terms of the computation speed both on synthetic 2D tasks and on well-recognized Wasserstein-2 benchmark.

The proposed regularized objective on the conjugate potentials relies on two additional hyperparameters, namely: the expectile coefficient $\tau$ and the expectile loss trade-off scaler $\lambda$, thus requiring a re-evaluation for new data. However, given the outcome of our ablation studies, the optimal parameters found on the Wasserstein-2 benchmark are *optimal enough* or at least provide a good starting point.

We believe that ENOT will become a new baseline to compare against for the future NOT solvers and will accelerate research in the applications of optimal transport in high-dimensional tasks, such as generative modelling. As for future directions, ENOT can be tested with the other types of cost functions, such as Lagrangian costs, defined on non-Euclidean spaces and in the dynamical optimal transport settings, such as flow matching.

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

# Appendix

## A  Expectile visualisations

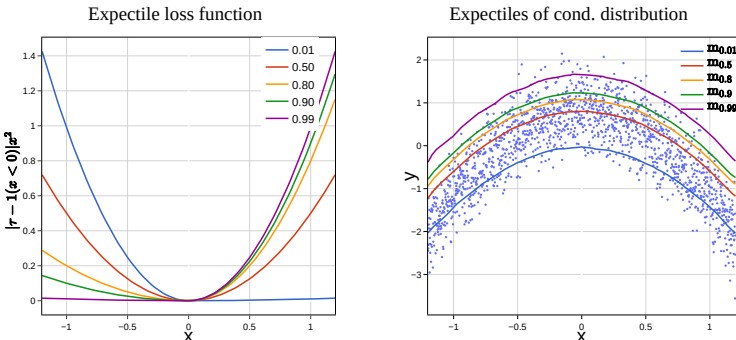

Figure 5: Expectile regression. **Left:** the asymmetric squared loss $L_\tau$. The value $\tau = 0.5$ corresponds to the standard MSE loss, while $\tau = 0.9$ and $\tau = 0.99$ give more weight to the positive differences. **Right:** expectile models $f_\tau(x)$. The value $\tau = 0.5$ corresponds to the conditional statistical mean of the distribution, and when $\tau \to 1$ it approximates the maximum operator over the corresponding values of $y$.

## B  Wasserstein-2 case.

For squared Euclidean cost $c(x, y) = \frac{1}{2}\|x - y\|^2$, one may use ordinary conjugation and replace the vector norms outside the supremum in (6). Let Kantorovich potential $g(y)$ equals $\frac{1}{2}\|y\|^2 - u(y)$, then

$$g^c(x) = \inf_y \left( \frac{1}{2}\|x - y\|^2 - \frac{1}{2}\|y\|^2 + u(y) \right) = \frac{1}{2}\|x\|^2 - u^*(x) \tag{21}$$

and consequently from (6) we derive that

$$\frac{1}{2}W_2(\boldsymbol{\alpha}, \boldsymbol{\beta}) = \frac{1}{2}\mathbb{E}_{\boldsymbol{\alpha}}\|x\|^2 + \frac{1}{2}\mathbb{E}_{\boldsymbol{\beta}}\|y\|^2 + \sup_{u \in L_1(\boldsymbol{\beta})}\Big[\mathbb{E}_{\boldsymbol{\alpha}}[-u^*(x)] + \mathbb{E}_{\boldsymbol{\beta}}[-u(y)]\Big]. \tag{22}$$

By equation (12) the corresponding optimal transport map $\widehat{T}(x)$ equals to the gradient of $\widehat{u}^*$ (argmaximum from the last formula):

$$\widehat{T}(x) = x - \nabla\widehat{f}(x) = \nabla\widehat{u}^*(x). \tag{23}$$

## C  Expanded Review of Related Work

In this section of the Appendix we briefly outline categorization of different approaches for estimating dual Kantorovich problem as proposed by (Amos [2023, 2022]), where the following differentiable amortization loss design choices are highlighted:

**A. Objective-based learning:** ($\mathcal{L}_{\text{amor}} = \mathcal{L}_{\text{obj}}$) methods utilize local information (4) to establish optimal descent direction for model's parameters $\theta$. (Dam et al. [2019]) predicts approximate amortized solution $T_\theta(x)$ from equation (7) by minimizing the next expression over minibatch of samples from $\boldsymbol{\alpha}$:

$$\mathcal{L}_{\text{obj}}(T_\theta(x)) = c(x, T_\theta(x)) - g_\eta(T_\theta(x)). \tag{24}$$

Methods max-min [MM] (Dam et al. [2019]), max-min batch-wise [MM-B] (Mallasto et al. [2019], Chen et al. [2019]), max-min + ICNN [MMv1] (Taghvaei and Jalali [2019]), Max-min + 2 ICNNs [MMv2] (Makkuva et al. [2020], Fan Jiaojiao and Chen [2021]), [W2OT-Objective] (Amos [2023]) use such objective-based amortization in order to learn optimal prediction. However, objective-based methods are limited by computational costs and predictions made by amortized models can be overestimated, resulting in suboptimal solution.

**B. Regression-based Amortization** ([Amos [2022]](#)) ($\mathcal{L}_{\text{amor}} = \mathcal{L}_{\text{reg}}$) is an instance of regression-based learning, which can be done by fitting model's prediction $T_\theta(x)$ into ground-truth solution $\widehat{T}(x)$, taking Euclidean distance as proximity measure:

$$\mathcal{L}_{\text{reg}}(T_\theta(x), \widehat{T}(x)) = \|T_\theta(x) - \widehat{T}(x)\|^2 \tag{25}$$

Such choice for learning $T_\theta(x)$ is computationally efficient and works best when ground-truth solutions $\widehat{T}(x)$ are provided. However, there are no guarantees for obtaining optimal solution when $\widehat{T}(x)$ is not unique.

**C. Cycle-based Amortization** ($\mathcal{L}_{\text{amor}} = \mathcal{L}_{\text{cycle}}$) is based on the first order optimality criteria for equation (4), i.e $\nabla_y c(x,y) = \nabla_y g_\eta(y)$. If $c(x,y) = \frac{1}{2}\|x - y\|^2$ then $\nabla_y c(x,y) = x - y$ and one may use the following expression in the loss

$$\min_\theta \mathbb{E}_{\boldsymbol{\alpha}} \mathcal{L}_{\text{cycle}}(T_\theta(x)) = \min_\theta \mathbb{E}_{\boldsymbol{\alpha}} \|x - T_\theta(x) - \nabla g_\eta(T_\theta(x))\|^2. \tag{26}$$

It is called *cycle-consistency* regularization. Method [W2] [Korotin et al. [2019]](#) uses this choice and substitutes it from the dual loss (9) to avoid solving max-min problem.

# D  Conjugate Function Approximation by Expectile

**Lemma D.1** ([Rudin [1976]](#)). *Let random vector $\xi$ have a compact support $\Omega$ and $\forall x \in \Omega$: $f_{n+1}(x) \geq f_n(x)$ be a sequence of continuous functions. Then from functional convergence $f_n \to f$ follows convergence of $f_n(\xi)$ to $f(\xi)$ with probability 1.*

**Theorem D.2.** *Denote by $f_\tau \in \mathcal{C}^0$ a non-parametric solution of expectile regression in class of continuous functions that approximates the $\tau$-th ($\tau > 0.5$) conditional expectile of c-conjugate transform $g(\eta) - c(\xi, \eta)$. Let function $g$ be upper-bounded and random vectors $\xi, \eta$ have compact support $\Omega$, then with probability 1*

$$\lim_{\tau \to 1} f_\tau(\xi) = -g^c(\xi) = \sup_{\eta \in \Omega} \{g(\eta) - c(\xi, \eta)\} \tag{27}$$

*Proof.* First note that $f_\tau(\xi) \leq -g^c(\xi)$ with probability 1, otherwise it would be possible to reduce the average value of the loss function $\mathcal{L}_\tau$ by taking $-g^c(\xi)$. By the monotonicity property of the expectile (ref. [Bellini et al. [2014]](#)) $\forall x \in \Omega, \tau_2 > \tau_1$: $f_{\tau_2}(x) > f_{\tau_1}(x)$. When $\tau \to 1$ for each $x \in \Omega$ it holds that $f_\tau(x)$ converges to $-g^c(x)$ as monotone and bounded sequence. By means of Lemma B.1 we also derive that $f_\tau(\xi)$ converges to $-g^c(\xi)$ with probability 1.

$\square$

# E  Implementation Details

## E.1  Environment and Libraries

We implement ENOT in [JAX](#) framework, making it fully compatible and easily integrable with the `OTT-JAX` library ([Cuturi et al. [2022]](#)). Moreover, since ENOT introduced expectile regularization, there is no additional overhead and whole procedure is easily *jit*-compiled, which is a drastic difference with previous approaches. To find the optimal hyperparameters in Appendix ([E.2](#)), we used [Weights & Biases](#) sweeps for hyperparameter grid search and [Hydra](#) for managing different setup configurations. ENOT implementation consists of only a single file, which is easy to reproduce and can be tested on other datasets of interest. We provide step-by-step tutorial of benchmarking ENOT by the following link [at OTT-JAX](#) or on the website `https://skylooop.github.io/enot/`.

## E.2  Hyperparameters for Wasserstein-2 Benchmark Tasks

Tables [4](#) and [6](#) provide detailed hyperparameter values used in the experiments in Section [5](#). To find the values of these parameters, we performed an extensive grid search sweep across different seeds, yielding the best results among the seeds, on average. We tried to be as close as possible in terms of hyperparameters to previous works. Likewise, we tested different choices of hidden layers

and found that the most stable training occurs at $n \geq 512$, but we found that for low-dimensional tasks (i.e $D \leq 64$), 128 neurons are enough to achieve lowest $\mathcal{L}_2^{\mathrm{UV}}$ compared to results reported in (Amos [2023], Makkuva et al. [2020], Korotin et al. [2023b]). Since ENOT does not introduce any additional computational overhead, increasing number of neurons will not slow down overall training time. Runtime comparison with (Amos [2023]) for W-2 benchmark presented in Table 9.

| Hyperparameter | Value |
|---|---|
| potential model $f_\theta$ | non-convex MLP |
| conjugate model $g_\theta$ | non-convex MLP |
| $f_\theta$ hidden layers | |
| $g_\theta$ hidden layers | [512, 512, 512] if $D \geq 64$, else [128, 128, 128] |
| # training iterations | 200 000 |
| activation function | ELU (Clevert et al. [2015]) |
| f optimizer | |
| g optimizer | Adam with cosine annealing ($\alpha = $ 1e-4) |
| Adam f $\beta$ | [0.9, 0.9] |
| Adam g $\beta$ | [0.9, 0.7] |
| initial learning rate | 3e-4 |
| expectile coef. $\lambda$ | 0.3 |
| expectile $\tau$ | 0.9 |
| batch size | 1024 |

Table 4: Hyperparameters for $D$-dimensional Gaussian Mixture Wasserstein-2 benchmark tasks.

| Hyperparameter | Value |
|---|---|
| potential model $f_\theta$ | non-convex MLP |
| conjugate model $g_\theta$ | non-convex MLP |
| $f_\theta$ hidden layers | |
| $g_\theta$ hidden layers | [64, 64, 64, 64] |
| # training iterations | 100 000 |
| activation function | ELU (Clevert et al. [2015]) |
| f optimizer | |
| g optimizer | Adam with cosine annealing ($\alpha = $ 1e-4) |
| Adam f $\beta$ | |
| Adam g $\beta$ | [0.9, 0.999] |
| initial learning rate | 5e-4 |
| expectile coef. $\lambda$ | 0.3 |
| expectile $\tau$ | 0.99 |
| batch size | 1024 |

Table 5: Hyperparameters for Synthetic 2D datasets from (Rout et al. [2021])

| Hyperparameter | Value |
|---|---|
| potential model $f_\theta$ | |
| conjugate model $g_\theta$ | ConvPotential (Amos [2023]) |
| hidden layers | 6 Conv Layers |
| # training iterations | 80 000 |
| activation function | ELU (Clevert et al. [2015]) |
| f optimizer | |
| g optimizer | Adam with cosine annealing ($\alpha = $ 1e-4) |
| Adam f $\beta$ | |
| Adam g $\beta$ | [0.5, 0.5] |
| initial learning rate | 3e-4 |
| expectile coef. $\lambda$ | 1.0 |
| expectile $\tau$ | 0.99 |
| batch size | 64 |

Table 6: Hyperparameters for CelebA64 Wasserstein-2 benchmark tasks.

| Hyperparameter | Value |
|---|---|
| potential model $f_\theta$ | UNet (Ronneberger et al. [2015]) |
| conjugate model $g_\theta$ | ResNet (Liu et al. [2019]) |
| # training iterations | 120 000 |
| activation function | ReLU and LeakyReLU(0.2) |
| f optimizer | |
| g optimizer | Adam with cosine annealing ($\alpha = $ 1e-2) |
| Adam f $\beta$ | |
| Adam g $\beta$ | [0.5, 0.5] |
| initial learning rate f | 1e-4 |
| initial learning rate g | 5e-5 |
| expectile coef. $\lambda$ | 1.0 |
| expectile $\tau$ | 0.98 |
| batch size | 64 |

Table 7: Hyperparameters for image to image translation tasks.

| MLP Hidden layers | Method | Runtime |
|---|---|---|
| $[64, 64, 64, 64]$ | W2OT (L-BFGS) | $\sim$ 60 min |
| | ENOT | $\sim$ 1.3 min |
| $[128, 128, 128, 128]$ | W2OT (L-BFGS) | $\sim$ 120 min |
| | ENOT | $\sim$ 1.3 min |
| $[256, 256, 256, 256]$ | W2OT (L-BFGS) | $\sim$ 300 min |
| | ENOT | $\sim$ 1.3 min |

Table 8: Runtime comparison for different layers sizes between W2OT (Amos [2023]) with default hyperparameters and ENOT on synthetic 2D data on tasks from Rout et al. [2021].

# F  Results on Synthetic 2D Datasets

Additionally, we evaluate the performance of ENOT on synthetic datasets, introduced in Makkuva et al. [2020] and Rout et al. [2021]. Here, all neural networks are initialized as non-convex MLPs, and for each optimal plan found by ENOT, we demonstrate difference between ground truth Sinkhorn $W_2(\boldsymbol{\alpha}, \boldsymbol{\beta})$ distance and optimal plan found by ENOT, which is recovered from learned potentials by equation (23). Figures 7 and 8 show the estimated optimal transport plans (in blue) both in forward and backward directions recovered by $T_{\theta\#}\boldsymbol{\alpha} \approx \boldsymbol{\beta}$ and $T^{-1}_{\eta\#}\boldsymbol{\beta} \approx \boldsymbol{\alpha}$ and the countour plots of the learned potential functions respectively. Also, in Table 8 we compare the runtime to complete 20k iterations using amortized method from W2OT (Amos [2023]). Table depicts how runtime changes for varying number of hidden layers in non-convex MLP, while keeping other hyperparameters for amortized model to those recommended from original paper with LBFGS solver. Additional details on the full list of hyperparameters is included in Appendix E.2 (Table 4).

## F.1  Synthetic 2D Tasks Details

Table 5 lists optimal parameters for ENOT for synthetic-2D tasks from Rout et al. [2021]. Amos [2023] pointed out that LeakyReLU activation works better compared to ELU used for Wasserstein-2 benchmark. However, for expectile regularisation we found out that ELU works as well for synthetic 2d tasks. We keep Adam $\beta$ parameters as default $[0.9, 0.999]$ and observe that 25k training iterations are enough to converge for tasks from Rout et al. [2021]. Moreover, we tried different neurons per layer and Table 8 shows runtime in minutes in comparison to previous state-of-the-art approach. Such speedups are made possible due to efficient utilization of `jit` compilation since ENOT does not use any inner optimizations.

## F.2  Additional results with varying expectile hyperparameter

To characterize ENOT performance as a function of expectile $\tau$, we performed evaluation with ranging it from $0.5$ to $0.999$ in several tasks:

| Method | $D=2$ | $D=4$ | $D=8$ | $D=16$ | $D=32$ | $D=64$ | $D=128$ | $D=256$ |
|---|---|---|---|---|---|---|---|---|
| W2OT | 157 | 108 | 91 | 140 | 246 | 397 | 571 | 1028 |
| **ENOT (Ours)** | 14 | 14 | 15 | 15 | 15 | 16 | 21 | 21 |

Table 9: Comparison of runtimes (in minutes) against the baseline (W2OT-Objective L-BFGS) on the high-dimensional (HD) tasks from the Wasserstein-2 benchmark with same networks architecture.

| $\tau$ | $\mathcal{L}_2^{\mathbf{UV}}$ ($D=256$) | $W_2$, Synth. 2D | FID (CelebA $\Rightarrow$ Anime) | MSE (CelebA $\Rightarrow$ Anime) |
|---|---|---|---|---|
| 0.5 | 0.55 | 33.47 | 16.43 | 0.264 |
| 0.6 | 0.52 | 12.63 | 16.28 | 0.260 |
| 0.7 | 0.51 | 9.59 | 15.95 | 0.265 |
| 0.8 | 0.49 | 1.4 | 15.19 | 0.262 |
| 0.9 | 0.5 | 0.06 | 13.87 | 0.266 |
| 0.95 | 0.54 | 0.03 | 14.27 | 0.267 |
| 0.999 | 0.55 | 0.02 | 13.91 | 0.288 |

Table 10: Performance of ENOT with varying levels of expectile hyperparameter $\tau$ on $W_2$ benchmark (**1st column**), showcasing intuition on convergence as $\tau \to 1$; Synthetic 2D data (**2nd column**); Image-to-Image translation FID (**3rd column**), and MSE (**4th column**).

- Image-to-image translation dataset (CelebA(f) to Anime with image size 64, Table 3);
- Wasserstein-2 benchmark with $D=256$ (Table 2);
- Synthetic 2D dataset from Figure 7.

In these experiments (Table 10), we observe a significant drop in performance when $\tau$ approaches 0.5 on the Synthetic 2D dataset and (CelebA(f) to Anime (in terms of FID). On Wasserstein-2 benchmark, the tendency is less evident. At the same time, values of $\tau$ in the range $[0.9, 1.0)$ always demonstrate convergence of ENOT, giving good results in all experiments. Setting $\tau = 1$ may cause an instability. This can be the case because, under certain conditions, the overall contribution of proposed regularization term will be zero, which means that the potentials can become unbounded. However, in our experiments, such an instability occurred extremely rarely (mostly due to bad optimizer parameters), resulting only in a slight drop in performance.

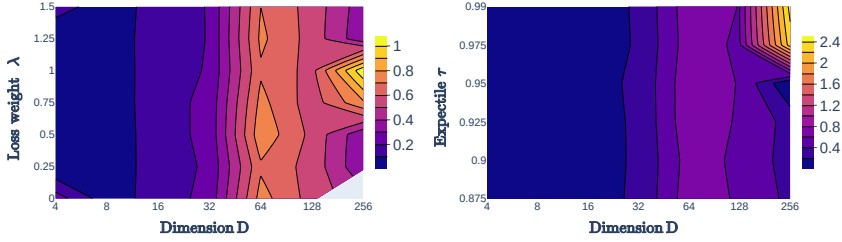

Figure 6: Ablating varying **Left**: loss weight $\lambda$ and **Right**: expectile $\tau$ coefficients in ENOT based on dimension of task from W2 benchmark. $\mathcal{L}_2^{\mathrm{UV}}$ is shown.

## F.3 Details on Generative Tasks

Table 7 provides details on hyperparameters used for unpaired image-to-image translation from Section 5.3. We observe that ENOT outperforms GAN based approaches, such as CycleGAN and StarGAN-v2. ENOT also outperforms the closest similiar recent approachs for generative modelling based on NOt such as Extremal OT (Gazdieva et al. [2023]) and Kernel OT (Korotin et al. [2023a]).

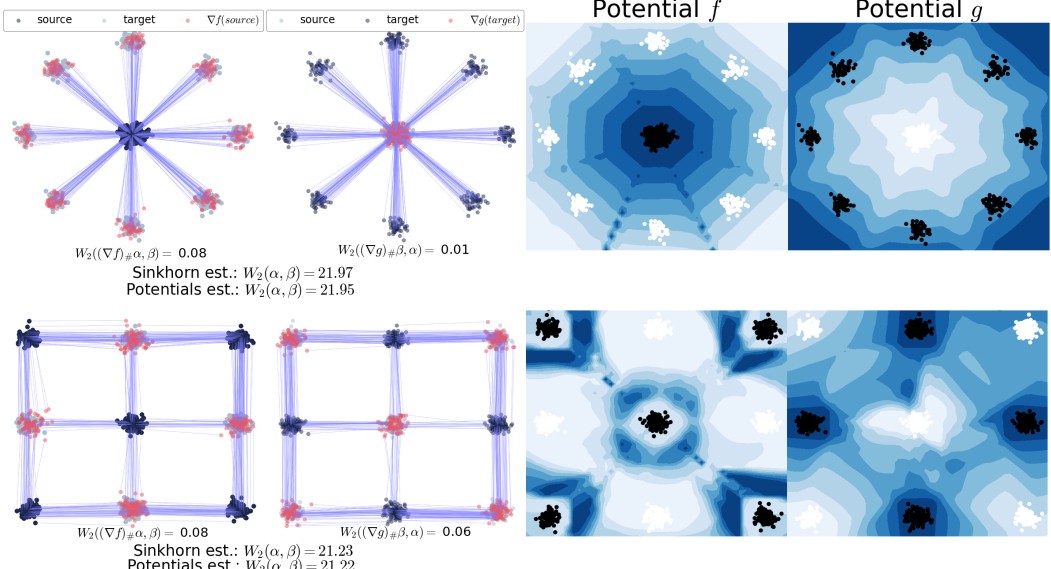

Figure 7: Recovered optimal transport plans ($\widehat{T}(x)$ and $\widehat{T}^{-1}(y)$ from (23)) and learned potentials contour plots obtained from solving OT dual problem (22) with squared Euclidean cost via ENOT regularisation on synthetic datasets from Makkuva et al. [2020]. Evaluation metric is Sinkhorn distance between the measures, i.e. $W_2(\widehat{T}_{\#}\boldsymbol{\alpha}, \boldsymbol{\beta})$, $W_2(\boldsymbol{\alpha}, \widehat{T}_{\#}^{-1}\boldsymbol{\beta})$. The estimated distance (22) from learned potentials compared with the reference value $W_2(\boldsymbol{\alpha}, \boldsymbol{\beta})$.

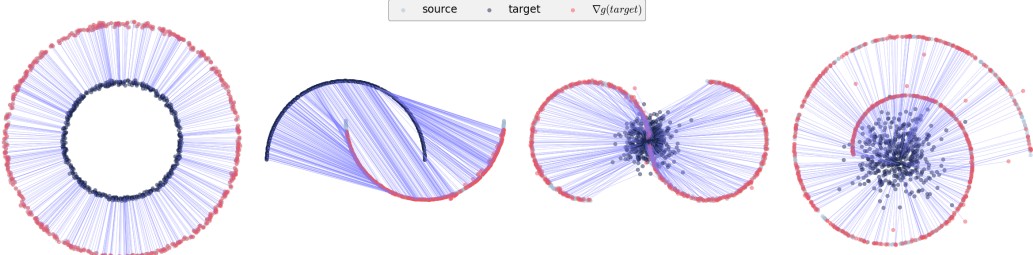

Figure 8: Recovered optimal transport push forward map (23) visualization for squared Euclidean cost using ENOT algorithm on synthetic datasets from Rout et al. [2021].

## F.4 W2-Benchmark Tasks Details

**High-dimensional measures (HD)** task from Korotin et al. [2019] tests whether OT solvers can redistribute mass among modes of varying measures. Different instantiations of Gaussian mixtures in dimensions $D$=2, 4, 16, ..., 256 are compared between each other via OT. In the benchmark, Mix3toMix10 is used, where source measure $\boldsymbol{\alpha}$ can consist of random mixture of 3 Gaussians and target measure consist of two random mixtures $\boldsymbol{\beta}_1, \boldsymbol{\beta}_2$ of 10 Gaussians. Afterwards, pretrained OT potentials $\nabla\psi_i\#\boldsymbol{\alpha} = \boldsymbol{\beta}$ are used to form the final pair as $(\boldsymbol{\alpha}, \frac{1}{2}(\nabla\psi_1 + \nabla\psi_2)\#\boldsymbol{\alpha})$.

**Images** task produces pair candidates for OT solvers in the form of high-dimensional images from CelebA64 faces dataset (Liu et al. [2015]). Different pretrained checkpoints (Early, Mid, Late, Final) from WGAN-QC model (Liu et al. [2019]) are used to pretrain potential models. Target measure for final checkpoint is constructed as average between learned potentials via [W2] solver and forms a pair input for OT algorithm as $(\boldsymbol{\alpha}_{\text{CelebA}}, \boldsymbol{\beta}_{Ckpt}) = (\boldsymbol{\alpha}_{\text{Final}}, [\frac{1}{2}(\nabla\psi^1 + \nabla\psi^2)\#\boldsymbol{\alpha}_{\text{Final}}])$. Figure 9 shows an example of pair of two such measures $(\boldsymbol{\alpha}, \boldsymbol{\beta})$.

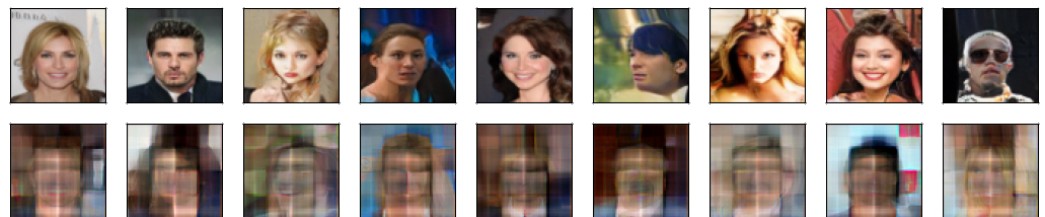

Figure 9: Example pair from W2-Benchmark of CelebA faces. **Top row**: Images, which were produced as final checkpoint from WGAN-QC model. **Bottom row**: Images, obtained from early checkpoint of WGAC-QC model.

## F.5 Unpaired Image to Image Additional Results

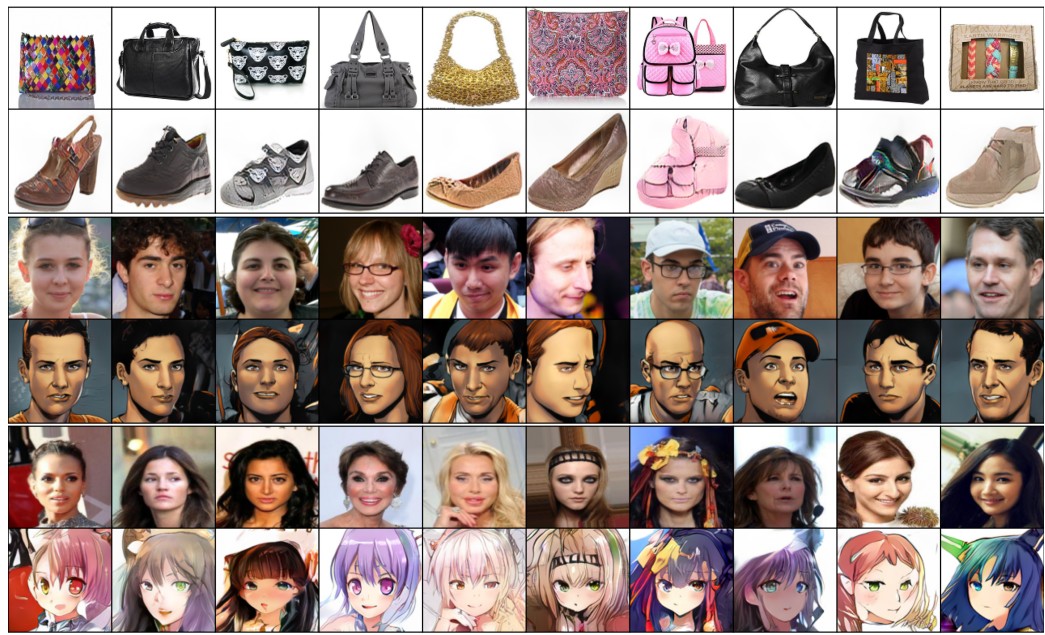

Figure 10: Optimal transportation mapping found by ENOT for **Top**: Handbag (top row) ⇒ Shoes (bottom row); **Middle**: FFHQ (top row) ⇒ Comics (bottom row); **Bottom**: CelebA(f) (top row) ⇒ Anime (bottom row) image-to-image translation tasks.

