# OpenReview forum: "Expectile Regularization for Fast and Accurate Training of Neural Optimal Transport"
_NeurIPS.cc/2024/Conference — NeurIPS 2024 spotlight_

### Official Review · Reviewer_qjtB · 2024-07-07

**Soundness:** 2
**Presentation:** 2
**Contribution:** 2
**Rating:** 6
**Confidence:** 3

**Summary:**

-	This paper introduces a new regularizer for learning Neural Optimal Transport. The proposed method, called Expectile-Regularized Neural Optimal Transport (ENOT), is based on the expectile regression. ENOT demonstrates competitive results on the Wasserstein-2 benchmark and Unparied Image-to-Image Translation.

**Strengths:**

- The proposed method is easy to implement as it only requires an additional regularizer.
- The ENOT achieves competitive results on the Wasserstein-2 benchmark.
- The ENOT proposes a new regularizer that softly guides the potential $g$ towards $g^{cc}$ using expectile regression.

**Weaknesses:**

- The derivation of the ENOT regularizer (Eq 17) requires further clarification.
- Please see the Questions Section below.

**Questions:**

- The ENOT and the Monge gap are similar in that they both serve as regularizers for inducing Neural Optimal Transport. Could you provide an additional quantitative comparison between ENOT and Monge gap? Currently, there are only qualitative examples in Fig 1 and 2.
- Table 3 presents the FID results on the Unpaired Image-to-Image Translation task. On the other hand, Table 10 in the appendix includes additional MSE results. For the Unpaired Image-to-Image Translation task, the goal of Neural Optimal Transport is to accurately approximate the target distribution (FID) while minimizing the cost function (MSE). Hence, the MSE results should be included in the main part of the manuscript.

**Typo:**
- Line 220: Table 10 -> 3
- Line 230: Figure 8 -> 4

**Limitations:**

-	The authors addressed the limitations of their work.

---

> ### Author Rebuttal · Authors · 2024-08-06
>
> **Q1: The ENOT and the Monge gap are similar in that they both serve as regularizers for inducing Neural Optimal Transport. Could you provide an additional quantitative comparison between ENOT and Monge gap? Currently, there are only qualitative examples in Fig 1 and 2.**
>
> We are thankful for this suggestion. We performed an additional comparison with the Monge Gap on $W_2$ benchmark (dataset from Table 2), which can be found below. Given the potential impact of these results on the ultimate algorithm selection for NOT, we propose to add the table to the main text.
> | Monge Gap    | ENOT         | DIM (W2 bench) |
> | :------------: | :------------: | :--------------: |
> | 0.1 +- 0.0   | 0.02 +- 0.0  | 2              |
> | 0.57 +- 0.0  | 0.03 +- 0.0  | 4              |
> | 2.05 +- 0.06 | 0.14 +- 0.01 | 8              |
> | 4.22 +- 0.1  | 0.24 +- 0.03 | 16             |
> | 7.24 +- 0.17 | 0.67 +- 0.02 | 32             |
> | 7.99 +- 0.19 | 0.56 +- 0.03 | 64             |
> | 9.1 +- 0.29  | 0.3 +- 0.01  | 128            |
> | 9.41 +- 0.21 | 0.51 +- 0.02 | 256            |
>
> **Q2: Table 3 presents the FID results on the Unpaired Image-to-Image Translation task. On the other hand, Table 10 in the appendix includes additional MSE results. For the Unpaired Image-to-Image Translation task, the goal of Neural Optimal Transport is to accurately approximate the target distribution (FID) while minimizing the cost function (MSE). Hence, the MSE results should be included in the main part of the manuscript.**
>
> We put the results with the MSE metric in the Appendix, because the majority of baselines we compare against did not include the outcome for those tasks. For convenience, we performed the experiments with the baseline methods and can include them into the main part of the paper:
>
> | Tasks (MSE)              | Extr. OT | Ker. OT | ENOT |
> | :---------------------: | :--------: | :-------: | :----: |
> | Handbags ⇒ Shoes 128  | 0.37     | 0.37    | 0.34 |
> | FFHQ ⇒ Comics 128     | 0.22     | 0.21\*  | 0.2  |
> | CelebA(f) ⇒ Anime 64  | 0.3      | 0.34\*  | 0.26 |
> | CelebA(f) ⇒ Anime 128 | 0.31\*   | 0.36    | 0.28 |
>
> Other issues:
> Equation (17): please kindly refer to our response to Reviewer E19z above.
>
> We are also thankful for spotting the typos, which we have corrected.

---

> > ### Comment · Reviewer_qjtB · 2024-08-11
> >
> > I appreciate the author for their clarifications and additional experiments. These have been helpful in addressing my concerns. Hence, I will raise my rating to 6.

---

> > > ### Author Response · Authors · 2024-08-12
> > >
> > > Thank you for raising the score and for helping us improve the paper.

---

### Official Review · Reviewer_QVdS · 2024-07-11

**Soundness:** 3
**Presentation:** 3
**Contribution:** 3
**Rating:** 7
**Confidence:** 3

**Summary:**

The paper introduces ENOT (Expectile-Regularized Neural Optimal Transport), a new method for training Neural Optimal Transport (NOT) models. It improves the efficiency of NOT solvers by incorporating a novel expectile regularization on dual Kantorovich potentials. Empirically, the authors use a Wasserstein-2 benchmark to demonstrate their method's improvement in accuracy and runtime.

**Strengths:**

- The paper is well-organized and clearly written.

- The proposed expectile regularization provides a new way to stabilize the learning of dual potentials. Empirically, this works really well, compared to existing NOT formulations.

- The experimental results are comprehensive.

- The authors were very upfront about the limitations of their approach, e.g., the requirement of extra hyperparameters $\tau$ and $\lambda$.

**Weaknesses:**

To my understanding, while the empirical performance of ENOT is strong, the technical novelty of the approach is limited. Expectile regularization combines the c-conjugate transform (7) and the quantile regression loss (15). ENOT merges two established ideas to produce strong empirical results.

**Questions:**

The quantile regularization term biases the estimation of the conjugate operator, making it non-exact. Despite this, the authors note that the regularization enables the method to outperform all exact methods. Could the authors speculate on the reasons for this? Is it solely due to the numerical stability of exact methods?

In Section 5.4, the authors mentioned

>Figure 8 presents the study of the impact of the proposed expectile regularization on the L_2^{\text{UV}} metric.

I believe this is a typo. The figure that the authors refer to is Figure 6 in the Appendix.

**Limitations:**

See "Weakness".

Nitpicking: the authors may want to consider differentiating the commands \citep and \citet in latex citation and use them accordingly.

---

> ### Author Rebuttal · Authors · 2024-08-06
>
> We are thankful for raising important questions and for the positive feedback.
>
> First, we would like to address the concerns about novelty:
>
> To the best of our knowledge, ENOT is the first approach that ventures into the approximation of the c-transform by means of expectile regularization. Moreover, ENOT is first to apply this approximation for efficient and accurate solution of Neural OT by completely mitigating additional $c$-conjugate optimization as done in prior works. These features provide at least a 3-fold better and a 10-fold faster training performance against the competitors.
>
> **Q1: The quantile regularization term biases the estimation of the conjugate operator, making it non-exact. Despite this, the authors note that the regularization enables the method to outperform all exact methods. Could the authors speculate on the reasons for this? Is it solely due to the numerical stability of exact methods?**
>
> We acknowledge the need to discuss this and would like to add the following text to the paper. In actuality, the expectile regularization does not introduce an additional bias, neither in theory, nor in practice.  The formal convergence for the c-conjugate transform, when $\tau$ converges to 1, is discussed in Appendix D.  At the end of training (or upon convergence), we obtain the exact estimate of the $c$-conjugate transformation. Other methods demand near-exact estimation at each optimization step, requiring additional inner optimization and introducing significant overhead. We assume that introduces an imbalance in the simultaneous optimization by $T$ and $g$ in Equation 6, underestimating the OT distance as a result.
>
> **Q2: Section 5.4 typo**
>
> Thank you for careful reading. We have made the correction.

---

> ### Comment · Reviewer_QVdS · 2024-08-11
> **Reponse to rebuttal**
>
> Thank you for your response. I especially appreciate your discussion on Q1. It would be great to see a similar discussion included in the paper.
>
> I've increased my score to 7. However, I must note that although I've used optimal transport in previous work, I don't consider myself an expert in the field. My support for the paper is mainly because of its strong empirical performance. For this reason, I recommend that the area chair give more weight to the opinions of OT experts when making the final decision.

---

> > ### Author Response · Authors · 2024-08-12
> >
> > Thank you for your feedback and for the final score. We are pleased to see you appreciate the empirical performance and are available for further conceptual explanations if needed during the next stage of the discussion.

---

### Official Review · Reviewer_ZPj5 · 2024-07-12

**Soundness:** 3
**Presentation:** 3
**Contribution:** 4
**Rating:** 7
**Confidence:** 4

**Summary:**

Authors provided a new, theoretically justified loss in the form of expectile regularisation which stabilize the learning of Neural Optimal Transport.  Importantly proposed method outperforms previous state-of-the-art approaches on the established Wasserstein-2 benchmark tasks and image-to-image by a large margin.

**Strengths:**

Originality:
- Expectile regularisation appears to be novel and not previously considered in OT literature.

Clarity:
- Paper is well written and easy to follow.

Significance:
- Paper providing an efficient non max-min solver for computing neural optimal transport maps, which is important.

**Weaknesses:**

Quality:
- Comparison with other methods is focusing mainly on Wasserstein-2 benchmark and image-to-image translation. Comparison with some popular in OT biology problems (see 1, 2) would improve the contribution.

References:
1) TrajectoryNet: A Dynamic Optimal Transport Network for Modeling Cellular Dynamics Alexander Tong, Jessie Huang, Guy Wolf, David van Dijk, Smita Krishnaswamy

2) Light Schrödinger Bridge: Alexander Korotin, Nikita Gushchin, Evgeny Burnaev

**Questions:**

- The authors consider two training modes depending on the ‘is_bidirectional’ parameter. In what cases the bidirectional training mode works better and what it depends on?
- The algorithm (Algorithm 1 line 2) states that in case 'is_bidirectional' is false $T(x)=\nabla f(x)$. From what it follows?
- Other algorithms that use expectile regression tend to use 0.9 expectile, why in your method you take the expectile parameter closer to 1?
- By what metric can we choose the hyperparameters of $\lambda$ and $\tau$ ?
- Partial OT is often required in practical tasks. It is possible to generalize this method to solve the problem of partial OT?
- L2 cost is probably not the best choice in the image-to-image translation task, can we use any cost, e.g. parameterised by the neural network, in coupling with your method?

**Limitations:**

Limitations are discussed.

---

> ### Author Rebuttal · Authors · 2024-08-06
>
> **Q1: Comparison with other methods is focusing mainly on Wasserstein-2 benchmark and image-to-image translation. Comparison with some popular in OT biology problems (see 1, 2) would improve the contribution.**
>
> We thank the reviewer for the links to these interesting articles and the application of OT in biology. We analyzed the cited works and concluded that our method is not explicitly suitable for solving the problems from these papers, because they require finding highly non-linear maps. But we have made a rough linear estimation on the proposed biological dataset (taken from TrajectoryNet paper) with our method, given in the table below. Despite learning linear maps, ENOT outperforms baseline methods based on score-matching (DSBM) and flow matching (SB-CFM).
> | Solver  | W1 metric    |
> | ------- | ------------ |
> | ENOT    | 0.88 +- 0.03   |
> | LightSB | 0.82 +- 0.01   |
> | SB-CFM  | 1.22 +- 0.4   |
> | DSBM    | 1.78 +- 0.42 |
>
> **Q2  The authors consider two training modes depending on the ‘is_bidirectional’ parameter. In what cases the bidirectional training mode works better and what it depends on?**
>
> The bidirectional mode works only with strongly convex functions $h(x-y)$. It significantly improves the solution when $T(x)$ is a discontinuous function (as in Figure 7). We propose to mention this in the Discussion Section.
>
> **Q3 The algorithm (Algorithm 1 line 2) states that in case 'is_bidirectional' is false . From what it follows?**
>
> When the parameter  is\_bidirectional = False, we can use any parameterization for $T(x)$ by a neural network. The expression $T= \nabla f$ is used to avoid introducing additional variables, but here, we assume **any** function.
>
> **Q4 Other algorithms that use expectile regression tend to use 0.9 expectile, why in your method you take the expectile parameter closer to 1?**
>
> The point is that with $\tau$ converging to 1, we get a more accurate estimate of the c-conjugate transformation. But from the experiments in Figure 4, we can conclude that the values in the range 0.9-1.0 also give a good result.
>
> **Q5 By what metric can we choose the hyperparameters**
>
> Conventionally, one could use L2-UVP metric for the experiments on W2 benchmark (Tables 1,2) and FID metric for the Image-to-Image tasks (Table 3).
>
> **Q6  Partial OT is often required in practical tasks. It is possible to generalize this method to solve the problem of partial OT?**
>
> Our method can be easily adapted for a special case of partial OT, when the whole source measure $\alpha$ and only a part of the target measure $\beta$ are used. Such problem is considered in Gazdieva et al. 2023 (citation from the paper).  For the general case, we are currently unable to suggest a proper method of adaptation. Still, if a new method for a neural partial OT is proposed elsewhere, it will also positively benefit from the use of our expectile regularization.
>
> **Q7 L2 cost is probably not the best choice in the image-to-image translation task, can we use any cost, e.g. parameterised by the neural network, in coupling with your method?**
>
> Indeed, the L2 distance between the activations from VGG16 layers can be used as a cost function. We also conducted experiments with it, but it turned out that the ordinary L2 cost gives more accurate results. Perhaps, this stems from the fact that the faces are centered in all images, enabling a direct comparison.

---

> > ### Comment · Reviewer_ZPj5 · 2024-08-10
> >
> > I appreciate the detailed answers and clarifications. I find this paper novel, easy to follow, and of interest to the community. The strengths of the paper significantly outweigh the weaknesses, and I increase my score.

---

> > > ### Author Response · Authors · 2024-08-12
> > >
> > > Thank you. We appreciate your suggestions and feedback.

---

### Official Review · Reviewer_E19z · 2024-07-13

**Soundness:** 3
**Presentation:** 2
**Contribution:** 3
**Rating:** 6
**Confidence:** 4

**Summary:**

This paper introduced a new framework for the training of Neural Optimal Transport, a recently emergent paradigm to enable the training of optimal transport plan in larger scale settings. The key contribution lies in the new and novel formulation of a regularized training loss that takes into account an expectile regularization term, which makes the max-min training of the generative NOT more stable. Empirical evaluations demonstrated improvments on both reducing the evaluated metrics and shortening the training runtime.

**Strengths:**

- The authors seem to have done a thorough literature review.

**Weaknesses:**

- Theoretical motivation unclear: lack of references on the instability of finding optimal c-conjugate (i.e. the motivation of this work). Where are the citations for the sentence from line 76 to 78? What does it mean to make the optimization problem more "balanced" with the expectile regression regularization on (line 86)? The same applies when the authors claim expectile regression is a popular option for estimatiing maximum of  distributions through neural networks (line 130).

- I suggest the authors rewrite their methodological section to be more compact and to the point (less verbose with the derivations, these can be put in the Appendix). Overall I was getting lost in this section, with the full main training objective never fully written, but have to resort to the pseudocode in algorithm 1 to see what was going on, and what the final loss was being used.


- **Major:** for eq (13) and eq (14) to hold you need both $f$ and $g$ to be convex functions (a la Brenier's theorem) with respect to the input; the only option is to use parameterization with ICNN. However, in table 4-5 and 7 the authors used non-convex MLP as parameterization; this is contradicting with the write up in methodology part. Could the authors elaborate on this?

- Ambiguous report of the empirical evaluations: most of the image task should use standard metric such as the FID. Many of the FIDs for the baselines are missing.

Writing and notation nitpick:

- I don't get the the derivation of the main loss function eq (17). The term in eq (15) is wrt to $\theta$ and a term $y - f_\theta$. What is the equivalence of $y$ and $f_\theta$ for $\mathcal{L}_\tau$ in equation (17) then?

- There should be explanation that we replace the quantity in Eq (7) to (5).

- Optimization and optimisation are used interchangeably. Stick to one of these.

- What is $\hat{\pi}$ right in the beginning of section 3? How are they different from $\pi$?. The same with $\hat{f}$ and $\hat{g}$. In the background section, all of the notations are added with hats $\hat{T}, \hat{\pi}, \hat{f}, \hat{g}$, then these when back to normal in Section 4; are they different problems from the introduction?

**Questions:**

See weaknesses.

**Limitations:**

See weaknesses.

---

> ### Author Rebuttal · Authors · 2024-08-06
>
> **Q1:  Lack of references on the instability of finding optimal c-conjugate (the motivation of this work). What does it mean to make the optimization problem more "balanced" with the expectile regression regularization on (line 86)? The same applies when the authors claim expectile regression is a popular option for estimating maximum of distributions through neural networks (line 130).**
>
> We are thankful for questions and for suggesting extra citations. We would like to stress that the main motivation behind our work was in finding fast, simple and accurate solver. The instability of finding accurate c-conjugate transform is the main bottleneck in the existing neural OT solvers. Those problems were thoroughly discussed in the following works (refer to Refs in our paper, e.g.,  Amos 2022a and Korotin et al 2021). For example, Korotin et al 2021 (Section 4.3, page 7) state that "The exact conjugate solver is slow since each optimization step solves a hard subproblem for computing the conjugate. Solvers that approximate the conjugate are also hard to optimize: they either diverge from the start or diverge after converging to nearly-optimal saddle point".
> Moreover this claim is empirically supported by the results in Tables 1 and 2 (i.e., W2OT-Objective, W2OT-Cycle diverged in all W2 tasks). Remarkably, our proposed regularization method stabilizes the overall training procedure.
>
> The term "more balanced" here means that we optimize optimal map $T$ and potential $g$ in the OT problem (Equation 6) synchronously and with the same frequency.
>
> Recently, expectile regression was used in some offline Reinforcement Learning algorithms and representation learning approaches. Among them, Implicit Q-Learning (IQL) methods [R1, R2].
>
> [R1] Ilya Kostrikov, Ashvin Nair and Sergey Levine. Offline Reinforcement Learning with Implicit Q-Learning, 2022.
>
> [R2] Dibya Ghosh, Chethan Anand Bhateja, Sergey Levine. Reinforcement Learning from Passive Data via Latent Intentions, 2023.
>
> **Q2 Major: for eq (13) and eq (14) to hold you need both f and g to be convex functions (a la Brenier's theorem) with respect to the input; the only option is to use parameterization with ICNN. However, in table 4-5 and 7 the authors used non-convex MLP as parameterization; this is contradicting with the write up in methodology part. Could the authors elaborate on this?**
>
> In Section (3), the variables with hats ($\hat{\pi}$, $\hat{T}$, $\hat{f}$, $\hat{g}$) correspond to the solution (argmins) of the OT problem (6).  The equations (13) and (14) hold under the specified conditions (source and target domains $X$, $Y$ are compact and the measures are absolutely continuous). So there is no strict requirement on $c$-concavity (or convexity); the potentials $f$ and $g$ must merely be the solution to the OT problem (ref. Santambrogio Theorem 1.17). During the training, we also may use the equations (13) and (14) with non-$c$-concave functions $f$ and $g$, because there are no restrictions on $T(x)$ in problem (6) and we can use any representation for the transport mapping function.
>
> Under weaker constraints (for example $X=Y=R^n$), the $c$-concavity of the potentials $f$ and $g$ may be required. In this case, we can rely on a local $c$-concavity in the data concentration region. In addition, if the conditions in equations (13) and (14) are not satisfied, we can use the mode (is\_bidirectional = False), in which we use an arbitrary function for $T(x)$ and do not express it through the potential.
>
> In practice, the is no benefit in using ICNN, because it makes sense only for the squared Euclidean cost, and even in this case the optimization problem deals with additional constraints, becoming more difficult. Similar empirical observations of under-performance of ICNNs were reported in (Amos 2022a).
>
> **Q3: Ambiguous report of the empirical evaluations: most of the image task should use standard metric such as the FID. Many of the FIDs for the baselines are missing.**
>
> Respectfully, the missing values in Table 3 mean that the corresponding experiments were not performed in the baseline publications. Given the suggestion, we performed additional experiments using the official implementation of the OT methods reported by the other authors to fill in the gaps, which we include in a table below and propose to add to our revision.
>
> | Task (FID)              | Extr. OT | Ker. OT | ENOT  |
> | :---------------------: | :--------: | :-------: | :-----: |
> | Handbags ⇒ Shoes 128  | 27.1     | 26.7    | 19.19 |
> | FFHQ ⇒ Comics 128     | 20.95    | 20.81\* | 17.11 |
> | CelebA(f) ⇒ Anime 64  | 14.65    | 18.28\* | 13.12 |
> | CelebA(f) ⇒ Anime 128 | 19.44\*  | 21.96   | 18.85 |
>
> **Q4: What are $\hat{\pi}, \hat{T}, \hat{f}, \hat{g}$?**
>
> We are sorry for the ambiguity. In Section 3, the hats ($\hat{\pi}, \hat{T}, \hat{f}, \hat{g}$) correspond to the solution of the OT problem in equation 6. As written in the beginning of this section, $\hat{\pi}$ is the **optimal** transport plan, $\hat{T}$ - corresponds to the **optimal** transport mapping,  $\hat{f}, \hat{g}$ are the **optimal** Kantorovich potentials. The notation without $\hat{\cdot}$ merely means that the optimization is done over the given variable.
>
> **Q5: Derivation of Equation 17:**
>
> We approximate the maximum of eq. (16) by $\tau$-expectile of $g^T(x) - c(x, y)$ conditioning on $y$. So the target (term $y$ in eq. 15) of the expectile regression here is  $g^T(x) - c(x, y)$. The model is then $-g_{\eta}(y)$ with a negative sign as we approximate c-transform of $g^T(x)$ as $\inf_x c(x, y) - g^T(x)$. The corresponding regression loss is: $
> L_{\tau} \big(
>      g^T(x) - c(x, y)  + g_\eta(y)
>     \big).$
> Taking definition of $g^T(x)$ from eq. (7), we derive the expression (17):
> $L_{\tau} \big(
>      c(x, T_\theta(x)) - g_\eta(T_\theta(x)) - c(x, y)  + g_\eta(y)
>     \big).$
>
> **Q6: Full Objective:** is defined in lines 168-169. The expression is $L_g(\eta) + L_f(\theta)  + \lambda R_g(\eta)$.

---

> > ### Comment · Reviewer_E19z · 2024-08-10
> >
> > I thank the authors for the detailed rebuttal.
> >
> > However, my major concern (Q2) regarding the soundness of this work remains:
> >
> > - I am fully understand Brenier's theorem only deals with squared Euclidean cost (W2 distance), but isn't this cost function was what you used in a major part of the experimental section anyway(Section 5.1 and Section 5.3)? This should mean c-concavity of $f$ and $g$ have to hold by enforcing it through the neural network parameterization?
> >
> > - Moreover, the L2 norm used in Section 5.2 (different cost functionals) violates the condition that h(x - y) must be strictly convex -- you can also verify this by yourself, or check Remark 1.18 in Santambrogio (2015) -- so I am not sure why your experimental results can hold all the case.

---

> ### Author Response · Authors · 2024-08-10
>
> **Q1**: Thank you for requesting this clarification. Yes, as indicated in Sections 5.1 and 5.3, and in most of our experiments, we used squared Euclidean cost. Under Brenier's theorem conditions for the squared Euclidean cost, it holds that $\hat{T}(x) = \nabla \hat{f} (x)$, where $\hat{f}$ is some convex function. It follows that the optimal potential $\hat{f}$ is convex, even if one uses non-convex potentials $f$ in the training process.
>
> Also, to reiterate our initial answer, there are no restrictions on $T(x)$ in Eq. (6) during the training and we can use any representation for this function, including $T(x) = \nabla f(x)$. Therefore, there is no need for $c$-concavity through the neural network parameterization whatsoever, and we do not use it, as it makes the solution worse in practice.
>
> The beauty is that our expectile regularization actually forces functions to be more $c$-concave according to the $c$-concavity criterion (refer to Villani et al. [2009] Proposition 5.8).
>
> &nbsp;
>
> **Q2**: We apologize for the ambiguity, which could have been avoided had we explicitly mentioned that we used flag `is_bidirectional=False` (meaning the training mode is one-directional in this example). Thank you for spotting the mismatch. So, in Section 5.2, we do not use (13) and (14) and we do not express $T(x)$ through the potentials, and the convexity of the cost function $h$ is not required. As such, we believe the issue should be resolved and there is no contradiction with Remark 1.18 in Santambrogio (2015). We will definitely add this additional clarification to the text.
>
> Please kindly notice that we do mention it in another part of the main paper (citations below). So, by adding this clarification to Section 5.2 also, we avoid any potential for the same confusion of the future readers of the paper. Thank you.
>
> > Lines 155-157 (Method section):  “The transport mapping $T_θ(x)$ has the same parameters as $f_θ(x)$ if it can be expressed through $f_θ$ (ref. 13), or otherwise, when $f_θ$ is not used (one-directional training), it is its own parameters.”
>
> > Lines 204-205 (Section 5.2): “we parametrize the map Tθ as a MLP”

---

> > ### Comment · Reviewer_E19z · 2024-08-11
> >
> > Thank you for the reply. Since my major concern is resolved, I'm raising the score towards acceptance. I would however expect the authors to better phrase their implementation of the experiments as they have promised in the rebuttal and response to my review.

---

> > > ### Author Response · Authors · 2024-08-12
> > >
> > > Thank you for your input and for raising the score. We commit to implement all the changes in the revised version of the paper.

---

### Official Review · Reviewer_2fef · 2024-07-21

**Soundness:** 3
**Presentation:** 3
**Contribution:** 3
**Rating:** 7
**Confidence:** 4

**Summary:**

This work tackles the problem of estimating the dual Kantorovich OT potentials
parametrized via neural networks.  They propose to approximate conjugate operator
using a well motivated loss called expectile regularization which approximates a conditional
maximum operator, well suited for estimating the c-transform. The authors illustrate the
performance of their method against several benchmarks showcasing a significant improvement
in both performance and runtime.

**Strengths:**

- Very simple, novel and intuitive idea with important and significant impact
- Thoroughly carried out experiments and evaluation

**Weaknesses:**

- Little analysis of the drawbacks of the proposed method as well as the effect of setting
the threhshold parameter tau (see Q1-2 below)

**Questions:**

1. The expectile loss is well suited to approximate the c-transform only if tau
is large enough (close to 1). Therefore, it would make sense to see a significant
drop in performance when tau is close to 0.5. It would be have been nice to validate
this intuition in figure 4 for instance.

2. I would expect this to lead to some form of tradeoff. Letting tau->1 should lead to
 some form of instability and/or slower convergence ? Is that the case ? How difficult
is this tradeoff to settle in practice ?

3. The fitted maps of Sinkhorn in fig1 and fig2 are odd, I suspect the entropic regularization
was very high, leading to a very very smooth transportation plan from which the fitted map
 was obtained by taking the maximum over values very close to each other.

- L74: is *squared
- L137: the*  proposed regularization approach
- L160: the* proposed

---

> ### Author Rebuttal · Authors · 2024-08-06
>
> **Q1: The expectile loss is well suited to approximate the c-transform only if tau is large enough (close to $1$). Therefore, it would make sense to see a significant drop in performance when tau is close to $0.5$. It would be have been nice to validate this intuition in figure 4 for instance.**
>
> Thank you for raising this question. To characterize ENOT performance as a function of expectile $\tau$, we performed additional evaluation with $\tau$ ranging from $0.5$ to $0.999$ on **1)** image-to-image translation dataset (CelebA(female) $\Rightarrow$ Anime with dim=$64$, Table $3$), **2)** $W_2$ benchmark with dim=$256$ (Table $2$), and **3)** 2D dataset from Figure $7$ (second row). In these experiments, we indeed observe a significant drop in performance when $\tau$ approaches $0.5$ ($0.7, 0.6, 0.5$) on the 2D dataset and CelebA (female) $\Rightarrow$ Anime with dim=$64$ (in terms of FID). On $W_2$ benchmark, the tendency is less evident. At the same time, values of $\tau$ in the range $\left[0.90, 1.0\right]$ always demonstrate convergence of ENOT, giving good results in all experiments. We will reflect on these new experiments in the revised manuscript.
>
> | Expectile ($\tau$) |  *$\mathcal{L}_2 ^\text{UV}$* (dim=$256$) | $W_2$ - 2D | FID (CelebA(f) -> Anime) | MSE (CelebA(f) -> Anime) |
> | :---------: | :----------------: | :--------------: | :------------------------:| :------------------------: |
> | 0.5       | 0.55             | 33.47          | 16.43                    | 0.264                    |
> | 0.6       | 0.52             | 12.63          | 16.28                    | 0.26                     |
> | 0.7       | 0.51             | 9.59           | 15.95                    | 0.265                    |
> | 0.8       | 0.49             | 1.4            | 15.19                    | 0.262                    |
> | 0.9       | 0.5              | 0.06           | 13.87                    | 0.266                    |
> | 0.95      | 0.54             | 0.03           | 14.27                    | 0.267                    |
> | 0.999     | 0.55             | 0.02           | 13.91                    | 0.288                    |
>
> **Q2: I would expect this to lead to some form of tradeoff. Letting tau->1 should lead to some form of instability and/or slower convergence ? Is that the case ? How difficult is this tradeoff to settle in practice ?**
>
> Setting $\tau \simeq 1$ may indeed cause an instability. This can be the case because, under certain conditions, the overall contribution of proposed regularization term will be zero, which means that the potentials can become unbounded. However, in our experiments, such an instability occurred extremely rarely (mostly due to bad optimizer parameters), resulting only in a slight drop in performance.  Yet, we carefully show that the optimal $\tau$ is usually in the range $\left[0.9, 1.0\right]$, with different values in this range yielding approximately the same result; so, selecting the optimal value of $\tau$ becomes simple.
>
> **Q3: The fitted maps of Sinkhorn in fig1 and fig2 are odd, I suspect the entropic regularization was very high, leading to a very very smooth transportation plan from which the fitted map was obtained by taking the maximum over values very close to each other.**
>
> Thank you for pointing this out. In order to be consistent with the experiments from the Monge gap paper (Uscidda and Cuturi [2023]), we used their exact default parameters of the entropic regularization, namely $\epsilon = 0.01$ $\cdot$ mean(C), with C being the cost matrix (in Figure 1, $\epsilon \sim 0.12)$, replicating the 'odd' look from the original paper. Moreover, we observe that lowering the values of $\epsilon$ produces plots of similar transportation smoothness.

---

### Author Rebuttal · Authors · 2024-08-06

We thank our reviewers for their constructive feedback. In the attached PDF, we provide additional experiments requested by the reviewers, showcasing the performance of ENOT. Corresponding tables are duplicated within individual responses to each reviewer.

---

### Decision · Program_Chairs · 2024-09-25

**Decision:**

Accept (spotlight)

**Comment:**

The paper introduces a new method for estimating the dual Kantorovich OT potentials, parameterized via neural networks, along with an interesting new loss function. It includes extensive benchmarking that demonstrates the practical benefits of the method, as well as a thorough literature review on the topic. Additionally, the paper proposes a new, efficient solver and is well-organized and clearly written. It also explores the limitations of the method and the impact of extra hyperparameters. While the motivation could be strengthened, and the reviewers have provided several suggestions for improvement in this direction, I recommend acceptance.